# Antiangiogenic Targets for Glioblastoma Therapy from a Pre-Clinical Approach, Using Nanoformulations

**DOI:** 10.3390/ijms21124490

**Published:** 2020-06-24

**Authors:** Gabriel Nery de Albuquerque Rego, Arielly da Hora Alves, Mariana Penteado Nucci, Javier Bustamante Mamani, Fernando Anselmo de Oliveira, Lionel Fernel Gamarra

**Affiliations:** 1Hospital Israelita Albert Einstein, São Paulo 05652-900, Brazil; gabriel.nery@einstein.br (G.N.d.A.R.); ariellydahora1997@gmail.com (A.d.H.A.); javierbm@einstein.br (J.B.M.); fernando.anselmo@einstein.br (F.A.d.O.); 2LIM44-Hospital das Clinicas HCFMUSP, Faculdade de Medicina, Universidade de São Paulo, São Paulo 01246-903, Brazil; mariana.nucci@hc.fm.usp.br

**Keywords:** glioblastoma, GBM, nanoparticle, nanomedicine, nanotherapy, angiogenesis, antiangiogenic therapy, tumor targeting

## Abstract

Glioblastoma (GBM) is the most aggressive tumor type whose resistance to conventional treatment is mediated, in part, by the angiogenic process. New treatments involving the application of nanoformulations composed of encapsulated drugs coupled to peptide motifs that direct drugs to specific targets triggered in angiogenesis have been developed to reach and modulate different phases of this process. We performed a systematic review with the search criterion (Glioblastoma OR Glioma) AND (Therapy OR Therapeutic) AND (Nanoparticle) AND (Antiangiogenic OR Angiogenesis OR Anti-angiogenic) in Pubmed, Scopus, and Cochrane databases, in which 312 articles were identified; of these, only 27 articles were included after selection and analysis of eligibility according to the inclusion and exclusion criteria. The data of the articles were analyzed in five contexts: the characteristics of the tumor cells; the animal models used to induce GBM for antiangiogenic treatment; the composition of nanoformulations and their physical and chemical characteristics; the therapeutic anti-angiogenic process; and methods for assessing the effects on antiangiogenic markers caused by therapies. The articles included in the review were heterogeneous and varied in practically all aspects related to nanoformulations and models. However, there was slight variance in the antiangiogenic effect analysis. CD31 was extensively used as a marker, which does not provide a view of the effects on the most diverse aspects involved in angiogenesis. Therefore, the present review highlighted the need for standardization between the different approaches of antiangiogenic therapy for the GBM model that allows a more effective meta-analysis and that helps in future translational studies.

## 1. Introduction

Glioblastoma (GBM) presents the worst prognosis for affected patients among all malignant tumors [1]. Regarding GBM treatment, advances in the pathophysiologic knowledge of disease in the last few decades have allowed the development and improvement of new therapeutic approaches in light of limited conventional therapeutics. Tumor resection followed by concomitant temozolomide administration and radiotherapy, in addition to occasional use of the bevacizumab (Beva) adjuvant, has not yet achieved a significant improvement in patient survival [2,3,4], with the exception of only a few cases with specific factors such as treatment of younger patients and maximal safe tumor resection, which has allowed survival after five years and corresponds to 5.6% of cases of GBM [5,6]. GBM has a complex character, showing high genetic heterogeneity with subclones existing in the same population of tumor cells [7].

The basic principle of antiangiogenic therapy is the disruption of blood supply to tumor cells, thus depriving the tumor of nutrients and oxygen and inhibiting the uncontrolled growth of tumor cells and their microenvironment [8,9,10,11].

When the tumor grows to a thickness of more than 2 mm, the cells in the nucleus are farther away from the blood supply, and the exposure of cancer cells to certain hypoxic stimuli [9,11] is the main trigger for initiating a series of reactions involving diverse molecular processes; for example, expression of hypoxia-inducible factors (HIF) that stimulate the expression of vascular endothelial growth factors (VEGFs) by tumor cells [8,9,12], in addition to communication with its auxiliary cells that stimulate the formation of other growth factors, such as platelet-derived growth factors (PDGFs) [12,13,14,15] and epidermal growth factors (EGFs) [10,16,17]. In chemotactic communication, inflammatory cells are recruited, such as tumor-associated macrophages (TAMs), which, in turn, stimulate a chain reaction of continuous tumor growth, intensifying the signaling of hypoxia to tumor cells, which further stimulates the expression of HIF and, mainly, VEGF [18,19,20,21,22]. All these growth factors in intercellular communication form several fronts of action that, in a dependent or independent manner, favor the formation of new blood vessels from pre-existing vessels [8,9,10]. Throughout the links of growth factors with the receptors present in endothelial cells that make up blood vessels, there is the activation of receptors, which are mostly transmembrane kinase proteins that, when activated, trigger a series of intracellular reactions to stimulate endothelial cell migration, proliferation, and differentiation as well as modulate vascular support cells [23,24,25]. For this migratory process to occur, it is necessary to have basement membrane degradation of the brain parenchyma extracellular matrix (ECM) through the stimulation of matrix metalloproteinases (MMPs) [11,26]. In all these processes of migration, proliferation, invasion, and remodeling, there is the ubiquitous action of calcium-dependent cell adhesion molecules, such as cadherins [27], or calcium-independent, such as integrins [28,29,30,31]. All of these described elements involved in angiogenesis are potential targets for therapeutic strategies, for example, the monoclonal antibody Beva that targets VEGF-A [32,33], or the tyrosine kinase inhibitor sorafenib (SFN) that targets the VEGF and PDGF receptors [34].

In order for these therapeutic agents to exert their antiangiogenic and antitumor effects, it is necessary that they penetrate the brain–blood barrier (BBB) [35,36]. One of the ways to penetrate the BBB is through local administration of therapeutic agents exercised by the technique of convection enhanced delivery (CED), in which a specific drug, through catheters connected with an infusion pump, is implanted directly into the developed tumor or in the parenchyma surrounding the tumor mass [37]. This technique is commonly applied in preclinical studies in GBM orthotopic models [38]; however, it has technical limitations and complications that make this type of therapy ineffective in clinical studies [39], requiring the development of structures that are safely administered in systemic circulation, are able to cross the BBB, and allow effective bioavailability of anti-tumor/anti-angiogenic drugs [35,36,40]. In this regard, a method that has been remarkable and promises to improve the delivery of bioactive compounds is the coupling of different types of nanostructured materials, such as a peptide based on polymeric structures [41,42], lipid-based liposomal formulations [43,44,45,46,47], and nanoshells [48]. These nanoformulations have the potential to provide clinically minimally invasive and targeted delivery of drugs with proven antiangiogenic effects or whose therapeutic potential is being pre-clinically tested [35,36,40].

One of the major scientific gaps in therapeutic strategies aimed at angiogenesis is how to effectively prevent the tumor from becoming nourished while also preventing GBM recurrence. It is necessary to highlight the actions of these nanoformulations on the various processes involved in angiogenesis, especially those that have multiple directions and assume different therapeutic front lines, so that they can interfere in a more comprehensive and effective way in the stages of angiogenesis, whether in triggering the cascade of hypoxia; in the release of growth factors; in dilation, permeability, proliferation, and migration of angiogenic blood vessels; in supporting cells; or also in the actions of metalloproteinases in the ECM, among others.

Therefore, the objective of the present work was to carry out a systematic review that focuses on the evaluation of the antiangiogenic therapeutic effects of different therapeutic strategies, having in common the use of nanoformulations for drug delivery in GBM orthotopic models developed in rodents.

## 2. Results

### 2.1. Study Selection

After applying search strategies in the databases, 312 articles were identified (113 articles of Pubmed, 175 of Scopus, and 24 of Cochrane Library). Following the Preferred Reporting Items for Systematic Reviews and Meta-Analyses (PRISMA) Guideline [49] and the inclusion and exclusion criteria established a priori, of 113 articles identified in Pubmed, 28 articles were excluded after screening (27 reviews and 1 written in another language) and 61 after eligibility analysis (21 articles reported no in vivo study analysis, 15 reported no intracranial glioma induction in the animal model, 15 reported no anti-angiogenic therapy, and 10 articles reported no angiogenesis evaluation after therapy), leaving only 24 articles included from this database. Of 175 articles identified in Scopus, 144 articles were excluded after screening (83 reviews, 1 article was written in a language other than English, 6 editorials or notes, 10 book chapters, 2 conference abstracts, and 42 duplicates in the search for PubMed) and 28 after eligibility analysis (10 reported no in vivo study analysis, 6 reported no intracranial glioma induction in an animal model, and 12 articles reported no angiogenesis evaluation after therapy), leaving only 3 articles included from this database. Of the 24 articles identified in the Cochrane Library, no articles were included, as 16 articles were reviews, 5 were protocols, and 3 were clinical articles in the screening analysis. In total, only 27 full-text articles not duplicated were included in this systematic review [32,34,38,41,43,44,45,48,50,51,52,53,54,55,56,57,58,59,60,61,62,63,64,65,66,67,68], as illustrated in Figure 1.

### 2.2. Cell Characteristics

The major tumoral cell characteristics used in the selected studies in our review are shown in Table 1. Of 27 selected studies for this review, 19 studies (70%) [32,34,38,43,45,48,52,53,55,56,57,58,59,62,63,64,65,66,67] used tumor cells from a human source, 6 (22%) [41,51,54,60,61,68] used tumor cell from rats, and 3 studies (11%) [44,50,66] used cells from mice. The study by Agemy [66] used human and mice tumor cell sources and also used lentivirus H-RasV12-shp53 to induce glioma cells. Regarding cell lineage of tumor cells from a human source, most studies (79%) [32,34,38,52,53,55,56,57,58,59,62,63,64,66,67] used the U87-MG cell lineage. For tumor cells from a rat source the C6 lineage was used in almost all studies (83%) [41,51,54,60,61], with the exception of Hekmatara [68], who used 101/8 cells produced initially by the injection of α-dimethylbenzanthracene into the brain Wistar rats; the GL261 lineage was used in two [44,50] of three studies where the tumor cells were from a mice source.

In 10 of the selected studies (37%) [32,41,48,52,55,56,57,58,64,66], modified cells were used to express some protein, the luciferase reporter protein being the most prevalent (70%) [32,48,55,56,57,58,66].

### 2.3. Tumoral Induction

The induction characteristics of the glioma model using the cells described above, as well as the animals, are described in Table 2. In all studies in this review, the tumor model was induced in rodents; mice were used in 21 studies (78%) [32,34,41,43,44,45,48,50,51,52,53,55,56,57,58,59,60,61,62,63,64,66,67] and rats were used in 6 studies (22%) [38,54,60,61,65,68].

In the review of studies that used mice as a model, the most used species was *BALB/C nude* [41,45,53,56,57,58,59,62] followed of *Nude athymic* [32,51,55,63,67] in eight (38%) and five (24%) studies, respectively; *C57BL/6* in two studies (10%) [44,50] and the other six studies used different species of mice such as *Mut6/cKO* [64], *Nonobese diabetic/severe combined immunodeficiency (NOD-SCID)* [66], *Swiss nude* [34], *CB17-SCID* [52], *ICR-Prkdc-SCID* [48], and *Rag2M* [43], representing 5% each. Mice used in the selected studies were male (43%) [41,44,45,48,58,59,62,63,64] or female (33%) [34,43,50,52,53,57,64], and the mice age ranged from 4 to 10 weeks in 12 studies (57%) [32,34,41,43,44,50,52,55,56,57,59,63].

Of the six studies that used rats as the model, the *Sprague–Dawley* species was used in 50% of these studies [54,60,61], two studies (33%) [38,65] used the *Nude athymic* species, and the study by Hekmatara [68] used *Wistar* rats. Rats were male in four studies (67%) [38,54,60,68], and only the study by Banerjee [61] used female rats. The minimum rat age was four weeks in the study by Hu [60], and the maximum age was 27 weeks in the study by Banerjee [61].

Regarding the cell quantity administered to rodents during glioma model induction, 11 of the selected studies (52%) [32,41,44,45,50,53,55,57,59,65,66] administered in mice more than 1 × 10^5^ cells to mice and in the other 8 studies (38%) [34,43,48,51,52,63,64,67] 1 × 10^5^ or fewer cells were administered. In rats, most studies (67%) [54,60,61,68] administered 1 × 10^6^, and in two studies (33%) [38,65] around 5 × 10^5^ cells was administered. The cells were administered using saline in 10 studies (37%) [34,38,51,55,57,59,60,62,63,66], while another 7 studies (26%) [45,48,50,52,53,58,65] used medium culture as the vehicle. The vehicle volume used in mice was 5 μL in 48% of the reported studies [32,34,41,50,51,52,53,59,62,63], and in rats, half the studies [38,54,60] administered 10 μL. In the study by Saito [38], 10 μL was used, divided in two doses of 5 μL.

Tumors were induced in all of the selected studies by the intracranial route, and in most studies, the tumor cells were implanted in the right cerebral hemisphere of rodents, with the exception of the studies by Sousa [32] and Lin [56] that used the left cerebral hemisphere. Specific brain regions were reported in the selected studies; in mice, 29% [34,41,51,52,59,62] implanted in the right striatum, followed by 5% each in the right frontal lobe [57], parenchyma [63], right caudate nucleus-putamen [43], right hippocampus [66], and right basal ganglia field [67]. In the study by Hekamtara [68], tumors were administered in the right lateral ventricle, and by Saito [38] in the striatum. 

### 2.4. Nanoparticles Used in Studies of Angiogenesis

The varied types of nanoparticles used in the selected studies for antiangiogenic processes in GBM treatment are shown in Table 3. Some studies reported the nanomaterial used depending on the application, including micelles [50], polymeric nanoparticles [32,41,51,57,62,67,68] lipid nanocapsules [34,52], liposomes [38,43,44,45], nanovesicles [64], nanoshells [48], nanobioconjugates [67], and superparamagnetic iron oxide nanoparticles (SPIONs) [53,65], among others [32,41,51,54,55,56,57,58,59,60,61,62,63,66,68]. Two drugs were mainly used in the therapeutic process: 15% of the selected studies used Paclitaxel (PTX) [56,59,61,62] and 15% of studies used Doxorubicin (DOX) [51,58,60,68]. Three studies (11%) [55,63,66] used peptides in nanoparticle formulations. The selected studies reported nanomaterial formulations of antibodies [52], plasmids [58], and proteins [64] as well as 78% associated with several drugs such as Luteolin (Lut) [50], Beva [32], SFN [34], Rhenium-188 [52], CARD-B6 constructed with 3 drugs: all-trans retinoic acid (ATRA), combretastatin A4 (CA4), and DOX [53], Docetaxel (DTX) [41,54], Camptothecin (CPT) [57], Epirubicin (EPR) [45], Chlorotoxin (CTX) [44], Irinotecan (IRN) [43], and Topotecan (TPT) [38], 52% of these drugs are derived from different types of plants [38,41,43,50,53,54,56,57,59,61,62], which is classified in 28% by taxanes PTX [56,59,61,62] and DTX [41,54] (semisynthetic analogue of PTX), 19% by natural alkaloid Vincristine [43], CPT [57], IRN [43], and TPT [38], 5% by combretastatin (CA-4) [53] dihydrostilbenoid derived from Combretum caffrum and 5% by flavonoid Lut [50] derived from Reseda luteola plant. However, three of the selected studies [54,63,65] reported on the antiangiogenic process for GBM therapy with a nanomaterial already used and well established in magnetic resonance imaging (MRI), photodynamic therapy (PDT), and tissue engineering areas with different approaches, including (i) the study by Janic [65], in which superparamagnetic nanoparticles based on iron oxide (commercially available as a contrast agent in MRI) were complexed with protamine sulfate; (ii) the study by Bechet [63], in which Chlorin was used in the nanoparticle formulations as a photosensitizer aiming PDT application; and (iii) the study by Xu [54], in which scaphoids were used, applied in tissue engineering, containing nanoparticles with DTX.

Most of the selected studies (85.2%) [32,34,41,44,45,48,50,51,52,53,54,55,56,58,59,60,61,62,63,64,66,67,68] synthetized nanomaterial, and 14.8% [38,43,57,65] used commercial nanoparticles. The main properties characterized by nanoparticles in the studies were size (hydrodynamic diameter), zeta potential (ζ), the polydispersity index (PDI), encapsulation efficiency (EE), and drug loading efficiency (DLE). The nanoparticle size in 96.3% of the selected studies [32,34,38,41,43,44,45,48,50,51,52,53,54,55,56,57,58,59,60,61,62,63,64,65,66,67] was smaller than 200 nm, and only the study by Hekmatara [68] used doxorubicin-nanoparticles (DOX-NP) with a size of 260 nm. The zeta potential in 92.59% of studies was negative, ranging from 0 to −57.9 mV [32,34,38,41,43,44,45,48,50,51,52,53,55,56,57,58,59,60,61,62,64,65,66,67,68]. The study by Bechet [63] that used PDT had a positive zeta potential (+22.6 mV and +42.9 mV), as did the study by Xu [54] that used a scaphoid scaffold adsorbed nanoparticles, with a zeta potential of +17.7 mV. PDI was lower than 0.3 in most of the studies [32,34,41,44,45,50,52,55,56,59,61,62,65,68], and this nanoparticle size characterization indicates low polydispersity. The EE varied in the studies according to the drug used. High EE was reported for SFN (105%) [34], TPT (>95%) [38], EPR (96.88–98.68%) [45], Lut (98.5%) [50], CTX coupled stable nucleic acid lipid particles (SNALPs) (85–95%) [44], PTX (88% and 86%) [61], Beva (82.47%) [32], and DOX (70% and 78.4%) [53,68]. Intermediate EE was reported between 44.84% and 56.33% as well as 44% and 53.24% for DOX [51,60] and PTX [56,62], respectively. The DLE in 26% of the studies [32,50,51,54,56,61,62] was lower than 5.18%, although two studies using CPT [57] and DOX [60] formulations showed high DLE values of 10% and 19.2%, respectively. The study by Lu [53] reported the highest DLE values (16.23% to 58.37%) in nanoparticle formulations involved with 11-ATRA, CA4, and DOX.

Regarding the release of drugs, 11 of the selected studies [32,34,45,50,51,53,54,56,59,60,62] performed this analysis. Seven of these studies [34,45,50,51,56,59,62] performed this evaluation using phosphate buffered saline (PBS), in which the study by Zhang [45] reported that the EPR release was lower than 2%; the studies with SFN [34] and Lut [50] reported releases of 11% and 46%, respectively; the assays with DOX [51] showed a release of 70% and 80%, and similar values were reported with the PTX assays [56,59,62], ranging from 69.25% to 79.4%. Acetate used in the study by Hu [60] produced a release of DOX of 50%. An assay performed for two nanoparticle formulations using rat plasma [62] achieved PTX releases of 82.91% and 84.53%. Other studies [32,53,54] did not report details on the release of drugs. 

### 2.5. Antiangiogenic Therapeutic Process for Glioblastoma

The therapeutic process of antiangiogenesis applied to the GBM models was analyzed in terms of the type of therapy, the therapeutic target, the route of drug administration, drug dose and the frequency used, the vehicle associated with the drug, and the time between induction of GBM and the outcome, focusing mainly on the efficiency in reducing the size of the tumor over time. The techniques applied for this evaluation are shown in Table 4.

The therapy type most reported (56%) in the selected studies for nanoformulation administration was the Drug Delivery system [32,41,43,45,50,51,53,54,56,58,59,60,61,62,68]. Three of these studies [45,58,61] specified as Targeted Drug Delivery, in which the nanoparticle was directed at a specific target molecule, and Dual-Targeting Drug Delivery was used in four studies [41,51,59,60]. This therapeutic type represents a new method of improving the action of anti-tumor drug delivery, exploiting the targets for various kinds of receptors expressed on the surface of or inside tumor cells.

Systemic therapy was used in 5 studies (18%) [55,57,64,66,67], in which functionalized nanoparticles with different molecules were used such as proteins, peptides, and tumor markers. CED is a method of direct delivery and was used in three (11%) of the selected studies [34,38,52].

PDT and Photothermal Therapy (PTT) modalities use photosensitizers to promote a therapeutic effect and were used in two (7%) studies [48,63]. The study by Bechet [63] used PDT, which, with the combined action of photosensitizer and visible light, resulted mainly in the formation of reactive oxygen species (ROS) and oxygen singlets (1O_2_). The study by Day [48] used PTT combined with the exposure of infrared light in the gold-based nanoparticles, which produced enough heat in the tumor region to induce cell death by protein denaturation and cell membrane rupture.

The study by Janic [65] applied a Ferumoxides-Protamine Sulfate (FePro) nanoparticulate contrast agent for cell therapy, and they reported the benefits of using endothelial progenitors cells (EPCs) in clinical applications as an alternative method to inhibit tumor vascular growth. The study by Costa [44] used a therapeutic approach based on indirect epigenetic modulation, using miR-21, which is associated with antiangiogenic chemotherapy and shows promising results.

The nanoformulations applied in the selected studies show modifications that favor improvement in the delivery of drugs to specific targets, appointing one or more process involved in the angiogenesis, as following: 4 studies [32,57,58,62] targeted the hypoxic cascade, 5 studies [34,48,55,59,63] to growth factor receptors, 5 studies [41,45,48,55,67] for adhesion molecules such as laminin and integrin, 6 studies [34,44,50,52,55,65] targeted to one of the signaling pathways of angiogenesis process, 3 studies [38,57,58] to the topoisomerase, 7 studies [51,52,60,61,65,66,68] tumor vasculature markers, and 8 studies [43,53,54,56,61,62,64,66] targeted unspecific markers.

Regarding analyses of the therapeutic process, the predominant route of administration of the nanoparticles carried in the drugs was the intravenous route, which was 78% of the selected studies [41,43,44,45,48,50,51,53,55,56,58,59,60,61,62,63,64,65,66,67,68]. Of these, 57% [41,44,45,48,50,51,56,60,63,64,68] specified that intravenous administration was via the caudal route, 15% of the selected studies [34,38,52,54] used the intratumoral route, and the study by Souza [32] used the intranasal route. Most of the selected studies (37%) [34,38,44,48,52,54,63,64,65,67] used a single dose, ranging from 10 µg/mouse [52] to 12 mg/Kg [64], with exception of two studies that used cell quantity as the dose, and the initiation of therapy ranged from 7 to 21 days [54] after tumor induction [67]. Then, therapy every 3 days was applied in 19% of the selected studies [41,45,59,60,62], using mainly 5 mg/Kg [59,60,62] (ranging from 0.1 [45] to 6 mg/Kg [41]) applied 2 [45,60] to 5 times [41,59,62] after 5 [60] to 14 days [45] of tumor induction. Other therapy frequency types were used in 11% of the selected studies, for example, weekly [32,43,57], every two days [55,56,58], every other day [51,53,66], and daily [50,61,68]. Therapy applied weekly [32,43,57] used doses ranging from 5 [32] to 21 mg/kg [43] mainly 2 [32,57] times after 4 [57] and 10 days [32] of tumor induction. Higher frequency was reported in the selected studies that applied therapy every other day [51,53,66] or every two days [55,56,58], in which the dose ranged from 50 µg/kg [58] to 10 mg/kg [51] applied 3 [58] to 12 times [66] after mainly 10 days [51,53,58,66] of tumor induction. In daily therapy [50,61,68], the dose ranged from 1.5 [68] to 50 mg/kg [50], applied 3 [68] to 14 times [61], starting in the very early tumor stage after 2 days of tumor induction [68].

Tumor reduction was the common end point reported in 43% of the selected studies [32,34,38,41,43,44,52,54,55,56,58,63,67,68], in which 54% of theses [52,54,55,56,67,68] reported high efficiency (more than 90% tumor reduction), 23% had intermediate efficiency [41,43,58] (between 70% to 80%), and 23% had low efficiency [32,44,63] (between 45% to 50%). This was evaluated mainly by imaging techniques (77%) such as bioluminescence (BLI) [32,55,56,58], fluorescence image (FLI) [54,55,59], MRI [34,52,53,54,63,65], positron emission tomography with computed tomography (PET/CT) [63], and laser speckle contrast analysis (LSCA) [53]; histologic analyses (33%) such as hematoxylin and eosin (H&E) [34,41,43,44,51,54,63,67,68] and Prussian blue [65]; and also other techniques such as survival curve [38,41,44,45,48,50,51,52,53,54,55,56,57,58,59,60,61,62,64,66], TUNEL assay [50,53,54,56,57,58,60], and Western blot [44,52,56] were also used.

### 2.6. Angiogenic Effects Evaluation

The effects of nanostructured materials on the neoangiogenic process in GBM tumors induced in animal models were commonly evaluated by VEGF [32,57,58,63,68,69], and CD31 [34,43,44,45,50,51,52,53,54,56,57,59,60,61,62,64,65,66], as shown in Table 5.

VEGF and its receptors are the most common angiogenic markers. Of the studies that evaluated VEGF expression 50% quantified decreased expression [32,57,58]. The studies by Kuang [58] and Souza [32] obtained similar results for the expression of endogenous VEGF determined by VEGF mRNA analysis by quantitative polymerase chain reaction (qPCR) (49–50%) after 9 and 10 days, respectively. Souza [32] also analyzed VEGF extracellular expression by ELISA, showing that modulation of VEGF caused by Beva-loaded Poly(d,l-lactic-co-glycolic) acid (PLGA) nanoparticles (NPs) at the intracellular level, although lower, had the desired effect of considerably inhibiting VEGF extracellular expression. In the study by Kuang [58], DGL-PEG-T7 [dendrigraft poly-l-lysines polyethylene glycol–peptide T7 (sequence His-Ala-Lle-Tyr-Pro-Arg-His)]/shVEGF could inhibit VEGF mRNA due to the T7 peptide that binds transferrin receptor (TfR) on the surface of the tumor cell, and the shVEGF subunit in the nanocomplex allowed DOX to inhibit tumor growth and angiogenesis.

Another classic endothelial cell marker for angiogenic blood vessels used in 19 of the selected studies [34,43,44,45,50,51,52,53,54,56,57,59,60,61,62,64,65,66] was CD31, in which 47% of the studies quantified results after therapy with different nanostructured materials applied [34,43,44,50,52,54,61,62]. The study by Wu [50] had the greatest reduction in CD31 expression with Lut with folic acid modified poly(ethylene glycol)-poly(e-caprolactone) (Lut/Fa-PEG-PCL) application (89% compared to the control group), significantly inhibiting the neovasculature of the GL261 tumor and playing an important role in inhibiting tumor cellular growth. A similar effect was observed in the study by Séhédic [52], with an 80% decrease in CD31 expression after seven days of lipid nanocapsules (LNC) (12G5-LNC^188^Re) application, in which clinical improvement was accompanied by locoregional effects on the tumor development including hypovascularization and stimulation of the recruitment of bone-marrow-derived TAM precursors (CD11b+ myeloid cells) or CD68-positive cells together with NOSII or Arg1 indicated the presence of pro- and anti-angiogenic macrophages, respectively, in which CD68 + / NOS-II was found inside towards the external part of the tumor, while CD68 +/Arg1 was exclusively present in the peripheral area of the tumor [52]. With the same therapeutic time (7 days), in the study by Feng [62], peptide (ACGLSGLGVA) with NP-PTX (CooP-NP-PTX) demonstrated great potential to improve anticancer activity and avoid the drawbacks of anti-angiogenic therapy alone with a reduction of 69% to 70% of CD31 expression. This reduction was also reported in the study by Bernarjee [61], after 15 days of application, the PTX-loaded solid lipid nanoparticles (SLN) modified with Tyr-3-octreotide (TOC) (PSM) demonstrated great potential as a dual target for tumor neovasculature and tumor cells due to TOC (in PSM surface) interaction with SSTR2 expressed in endothelial cells of the neovasculature-improving the PTX antiangiogenic effects. The study by Xu [54] also reported a similar reduction in CD31 expression, although 21 days after the end of treatment, the DTX-NPs-dBECM (decellularized brain extracellular matrix) complex displayed effective anti-angiogenic effects. With less efficiency in the ability to reduce CD31 expression (19–21%), the study by Clavreul [34] showed that lipic nanocapsules with SFN (SFN-LNCs) decreased the proportion of proliferating cells and the tumor vessel area 7 days after the end of treatment, inducing an early increase in tumor blood flow and vascular normalization process. The study of Varreault [43] also showed reduced tumoral blood vessel density 21 days after the end of treatment, suggesting restoration of the vessel architecture to a more normal state.

Regarding the technical evaluation, of the 25 studies that used immunoassays by marking angiogenic epitopes present in tumor tissue [34,38,41,43,44,45,50,51,52,53,54,55,56,57,58,59,60,61,62,63,64,65,66,67,68], most (64%) markings were made using the immunohistochemistry (IHC) technique [38,44,50,52,53,54,55,57,60,61,63,65,67,68], while in 44%, immunofluorescence (IF) was performed [34,41,43,45,51,56,58,59,62,64,66].

IHC was used for angiogenic labeling in 14 studies (64%) [38,44,50,52,53,54,55,57,60,61,63,65,67,68]; 9 (64%) used the CD31 protein as the main marker [44,50,52,53,54,57,60,61,65]; 3 studies (21%) used VEGF [57,63,68]; and 2 (14%) detected laminin [38,67]. In addition to CD31, another marker was used to detect endothelial cells, CD34 [55,58]. The study by Wang [55] associated CD34 labeling in conjunction with staining tissue with Periodic Acid-Schiff to differentiate endothelial-lined vessels and vasculogenic mimicry. Other markers detected were CD11b [52], MMP9 [52], von Willebrand Factor (vWF) [65], isolection B4 [68], CA IX [57], and CD68 [52].

IF analysis of tissue detection also predominantly used CD31 (81.8%) as the chosen epitope [34,43,45,51,56,59,62,64,66]. The study by Verreault [43] used double labeling for detection of CD31 and Collagen IV. Other tissue markers recognized by secondary fluorescent antibodies were used to evaluate functional blood vessels, such as HIFα1 [41] and lectin [58].

Another immunoassay reported in the selected studies was Western Blot (WB). The study by Lin [56] used WB to detect the presence of SPARC and gp60, two binding proteins of albumin that are shown to be expressed in blood vessels. The study by Saito [38] established the expression of blood vessels by laminin labeling and recognized the inhibition of Akt protein phosphorylation, which is related to the antiangiogenic activity. In addition to this, liposomal TPT decreased the expression of both proteins when compared to free TPT [38]. The study by Lin [57] reported the detection of VEGF expression by WB when performing a tissue analysis, despite having detected CD31 and CA IX by IHC. The ELISA immunoassay was reported in only one study [32] to detect VEGF protein levels.

The four studies [34,43,48,53] that evaluated angiogenesis by image techniques all evaluated blood perfusion in different ways. The laser speckle contrast images (LSCIs) recorded a 58% decrease in vascular permeability/flow after treatment with CARD-B6 (nanoparticles with B6 loading three drugs) [53]. MRI perfusion recorded a 24% decrease in blood flow due to the action of SFN-LNC [34], similar to the application of Dynamic Contrast-Enhanced-MRI. This made it possible to calculate the 85% reduction in Ktrans, which expresses a transfer volume constant between the vasculature and tissue compartment, by the action of Irinophore C^TM^ [43].

In front of all results obtained in the systematic review, Figure 2 schematically shown the antiangiogenic nanoformulations for glioblastoma therapy from a pre-clinical approach and the angiogenic process in the tumor microenvironment, as well as the pie charts of the main results found in the systematic review, and the quantification of antiangiogenic efficiency appointed by each study.

## 3. Discussion

In this systematic review, we showed different antiangiogenic therapies using nanoformulations in the GBM model. In the analysis of antiangiogenic therapy, the tumor cell type was considered as well as the GBM animal model, the characteristics of nanoformulations to reach therapeutic targets, and how they interfered with the angiogenic process.

An adequate GBM animal model requires a genetic pattern that resembles human GBM; has genetic, epigenetic, and phenotypic intratumoral heterogeneity; an adequate microenvironment in relation to immunocompetence; presence of the blood–brain barrier (BBB); and interactions between tumor cells and healthy cells that are reproducible [70]. Therefore, many of the selected studies used human U87-MG tumor cells [32,34,38,52,53,55,56,57,58,59,62,63,64,66,67], in which the tumor has a diffusely invasive infiltration pattern into normal brain parenchyma, resistance to therapy, and high recurrence rate [70]. Studies also used rat C6 tumor cells [41,51,54,60,61], in which the tumor mimics several features of human GBM including a high mitotic index, focal tumor necrosis, parenchymal invasion, and neoangiogenesis [71,72]. The GBM orthotopic model was induced mainly in immunosuppressed animals (e.g., *nude* or *SCID*) because this immunodeficient condition allows a greater possibility of the growth of human (xenogeneic) tumor cells after implantation, preventing their rejection [73]. GBM subcutaneous model is widely used in the literature due to its technical ease and high productivity, but it is not recommended for use in exploring the brain’s infiltrative behavior and lacks adequate brain microenvironments, which are important factors in the study of angiogenesis [74].

In order to improve the efficiency of antiangiogenic therapy applied to the GBM model, nanoformulations were developed to target the different processes involved in angiogenesis. The hypoxic cascade is the main driver of sprouting angiogenesis through the expression of HIF-1, a heterodimer composed of HIF-1α and HIF-1β. Whereas HIF-1β is constitutively expressed, the level of HIF-1α is low under normoxic conditions [11]. Due to hypoxia, increased HIF-1α expression results in the increase of mRNA VEGF and sequential expression of VEGF in GBM tumor cells [8,9,12]. Only the study by Lin [57] used this first stage of the angiogenic process as a therapeutic target, interfering in the expressions of HIF-1α, CA IX, and VEGF in the hypoxia cascade. The study by Gao [41] evaluated the antiangiogenic process and detected the accumulation of HIF-1α expression in the GBM orthotopic model induced in *BALB/c nude mice* with C6 cells, as an angiogenic marker. Increased HIF-1α expression induces CA IX expression, a promissory endogenous hypoxia-related cell surface enzyme [75,76]. There is a need for tumor cells to adapt to the highly acidic extracellular microenvironment caused by increased metabolic production of CO_2_ and lactic acid, in which CA IX catalytic activity is inhibited by low pH and is half-maximal (i.e., the pK) at pH~6.8 [75,76]. The expression of CA IX in GBM cases should be further investigated since its expression has shown to be more correlated with the prognosis of esophageal and gastric adenocarcinomas than the expression of VEGF [76]. Another protein upregulated by HIF-1α is CXCR4, a chemokine receptor for SDF-1α also known as CXCL12, whose expression occurs in hypoxic tumoral microenvironment conditions and vascular ischemia [69,77]. The studies by Séhédic [52] and Janic [65] used the CXCRL12 (SDF-1α) chemokine receptor, i.e., CXCR4, as their main target, which activates different signaling pathways including phosphatidylinositol-3 kinase (PI3K)/Akt and MAP-kinases [78]. In adults, the SDF-1/CXCR4 axis plays a role in GBM development, tumor cell proliferation, and invasiveness via activation of matrix metalloproteinases (MMPs) [78]. The study by Janic [65] applied cell therapy with umbilical cord EPCs carrying the magnetic agent Ferumoxides-Protamine Sulfate. EPCs have neovascularization potential due to the attraction to CXCR4 receptors, resulting from the hypoxia process present in the tumor microenvironment, activating this signaling pathway [79,80,81]. Thus, it is relevant to use EPCs labeled with magnetic nanoparticles as MRI probes for monitoring the neovascularization process [65]. In septic patients, EPC expression of CXCR-4 and high serum concentrations of VEGF, SDF-1α, and Ang-2 were associated with probability of survival, showing that the SDF-1/CXCR4 axis plays a crucial role in homing EPCs in the course of sepsis [80].

VEGF expression is the main product of hypoxia related to the angiogenesis process, which was approached in some studies [32,57] as the therapeutic target. The study by Souza [32] used Beva, a well-established monoclonal antibody that acts by directly inhibiting extracellular VEGF, linked in polymeric nanoparticles [(Poly(d,l-lactic-co-glycolic) acid] to improve the antibody-based therapeutic delivery and reduce off-target toxicity, a strategy that resulted in a greater (around 50%) antiangiogenic effect analyzed by VEGF protein and endogenous mRNA VEGF. These polymeric nanoparticles are used as carriers for different drugs such as in chemotherapy, which are either adsorbed on the surface or encapsulated within the nanoparticles. These improve the release kinetics, the compatibility with some active agents, have no oxidation issues (as with phospholipids), and enhance shelf-life [82]. Other studies [63,68] also detected VEGF in antiangiogenic therapy but did not quantify the protein expression after therapy. The study by Bechet [63] showed an intense decrease of Ki67 and VEGF protein expressions in the tumor tissue immediately after interstitial photodynamic therapy (iPDT), and the study by Hekmatara [68] showed that DOX in solution led to a slight decrease in necrosis and microvascular proliferation, whereas DOX bound to polysorbate 80-coated poly(butyl cyanoacrylate) nanoparticles drastically decreases necrosis and led to the complete disappearance of microvascular proliferation. DOX is an antibiotic applied as chemotherapeutic medication and is synthesized by *Streptomyces peucetius,* which inhibits topoisomerase II (topo II) through multifactorial mechanisms involved in the cytotoxic response. It is clinically effective in the management of hematological malignancies and solid tumors [83]. This antibiotic was used in some of the selected studies, but it was linked with different formulations, such as in polylactic acid nanoparticles linked with the CRKRLDRNC peptide to direct the nanoformulation to IL-4R [51]; with shVEGF, an anti-VEGF gene drug [58]; and with polydopamine-coated mesoporous silica functionalized with Asn-Gly-Arg (NGR) targeting CD13 [59], which is expressed exclusively on the angiogenic endothelium and not on normal vasculature [84].

VEGF angiogenic effects are mediated by three receptor tyrosine kinases (RTKs): VEGFR-1 and VEGFR-2, which play major roles in physiological as well as pathological angiogenesis, including tumor angiogenesis, and VEGFR-3, which can regulate angiogenesis in early embryogenesis but mostly functions as a critical regulator of lymphangiogenesis [85]. These VEGF receptors were used as therapeutic targets in three of the selected studies [34,48,63]. Using nanoformulations for external therapy, the study by Day [48] targeted VEGF linked to nanoshells and applied PTT to achieve an effect in the endothelial cell. They showed a decrease of 42% vessel density, evaluated by intravital microscopy after 3 days of treatment, as well as disruption of tumor vessels, predominantly in the tumor and at its periphery, but not in the adjacent normal brain. The study by Béchet [63] administered the photosensitizer chlorin conjugate with a heptapeptide (ATWLPPR), specific for the VEGFR and its membrane-bound coreceptor neuropilin-1 (NRP-1), for iPDT; they showed vascular disruption, edema in both the tumor and brain-adjacent tumor areas, and also a large decrease in VEGF expression. The heptapeptide (ATWLPPR) targeting the NRP-1 protein was also used in the study by Hu [59], in which this peptide was associated with CGKRK forming the dual-decorated nanoparticulate (designated AC-NP) to achieve a dual-targeting effect for angiogenic blood vessels and the tumor microenvironment. Targeting RTKs receptors, the study by Clavreul [34] used SFN conjugated with LNCs applied locally through CED. This technique associated with LNCs has the advantage of bypassing the BBB and results in a greater volume of disponible drug, meaning promising results were yielded in orthotopic GBM models [52,86]. This strategy is also commonly used with PTX [87] as well as anti-EGFR and anti-Galectin-1 siRNAs [88]. SFN is a multikinase inhibitor chemotherapy agent that acts on endothelial cell-surface RTKs (VEGFR-2, VEGRF-3, PDGFR-β, c-kit, and Flt-3). Recently, novel cross-family interactions between VEGF, PDGF, and their receptors were discovered, proposing a new mechanism for understanding anti-angiogenic drug resistance, and this presents an expanded role of growth factor signaling with significance in health and disease [89]. The structural difference in both RTKs is that PDGFRs have five extracellular immunoglobulin-like (IgG-like) domains [90] and VEGFRs have seven extracellular IgG-like domains [91].

The abundant secretion of VEGF by tumor cells promotes the formation of poorly developed blood vessels and with inadequate coating of endothelial supporting cells, i.e., the pericytes. Transforming the conditions of the blood vessels associated with the tumor into vessels properly covered with perycites is one of the main objectives of the application of nanoformulations and another therapies applied to combat aggressive tumors that favor angiogenesis. Goel [92] theorized the vascular normalization process, in which the restoration of normal vasculature condition is due to the greater effectiveness of antiangiogenic factors rather than proangiogenic factors. This normalized vasculature would result in increasing tumor perfusion, which would cause a reduction and consequent absence of hypoxia conditions. Despite the importance of pericytes in the process of vascular normalization, none of the selected studies analyzed the influence of the respective therapies on these cells. However, some therapies analyzed, such as the application of CARD-B6 [53], associated their therapeutic success with the reduction of blood flow. Likewise, the suggested restoration of vessel architecture to a more normal state provoked by I Irinophore C^TM^ was associated with reduced blood flow expressed by Ktrans values [43]. Blocking VEGFR2, which can be achieved with SFN [34] and through DC101, a VEGFR-2 monoclonal antibody, promotes Ang-1/Tie2 signaling, which correlates with the recruitment of pericytes through blood vessels [93]. The relatively narrow normalization window depends on the dosage of VEGF inhibitors, that varies with tumor type, schedule and the signaling inhibitor [8,94]. It is important that the process of VEGF inhibition don’t be excessive, but that a balance be achieved between the proangiogenic and antiangiogenic factors, since the total inhibition of VEGF signals represents a hypoxic picture for tumor cells, favoring to a risk of metastatic spread [95]. In addition, vascular normalization would promote decreased permeability of proangiogenic factors by tightening cell–cell junctions and decreasing cell adhesion of endothelial cells to extracellular tumor matrix and adjacent cells mediated by signaling of integrins [8,96].

Integrins are key regulators of communication between cells and their microenvironment. They are directly involved in the interplay between pro-angiogenic growth factors and their receptors, thereby exercising varied biological functions such as cell adhesion, migration/invasion, proliferation, survival, and apoptosis [11,97]. Integrins are a set of heterodimeric transmembrane glycoproteins made up of different α and β subunits, and αvβ3-Integrin has received much attention as a potential anti-angiogenic target because it is upregulated in tumor-associated blood vessels [11]. Agents targeting αvβ3-integrin are now showing some success in phase III clinical trials for the treatment of GBM, but the exact function of this integrin in tumor angiogenesis is still relatively unknown [98]. αvβ3-Integrin recognizes the Arg-Gly-Asp (RGD) peptide present in ECM proteins such as fibronectin, vitronectin, von Willebrand factor, osteopontin, and laminin, contributing to information exchange between the intracellular molecules and ECM proteins [11]. The synthesis of peptides and peptidomimetics coupled to RGD improved drug delivery to tumor vessels in varied studies [41,45,55] that sought to target antitumor agents to heparin [55], EPR [45], and DTX [41], and different types of encapsulation were also developed. The study by Wang [55] used a heparin-based polymer conjugated with cRGD and SWLAYPGAVSYR (SWL; S = serine, W = tryptophan, L = leucine, A = alanine, Y = tyrosine, P = proline, and V = valine) peptides to promote selective affinity for the overexpressed integrin and EphA2 tyrosine kinase receptor, respectively, reaching 56% simultaneous inhibition of endothelial-lined blood vessels and 82% vasculogenic mimicry. VEGFR-2 plays a key role in vasculogenic mimicry formation, neovascularization, and tumor initiation by glioma stem-like cells [99], and EphA2 and its receptors are significantly involved in blood vessel formation and remodeling during the vascular development of cancers [25]. EphA2 may regulate vessel sprouting during developmental angiogenesis independently via inhibition at both the gene and protein levels of VEGFR-2, without affecting VEGF expression by GBM cells [25]. The study by Zhang [45] modified the surface of liposomes with a thiolated cyclic pentapeptide derivative containing RGD conjugated with DSPE-PEG2000-maleimide, aiming at a nanoformulation capable of crossing the BBB and target GBM neovasculature. The study by Gao [41] developed a nanoformulation with interleukin-13 and RGD to target DTX in GBM neovasculature; this nanoformulation prevented HIF1α accumulation in the GBM site. Another study [48] used αvβ3-Integrin as the therapeutic target but did not specify the RGD application. Stimulation of proliferation and endothelial cell migration in the angiogenic process, when mediated by VEGF, occurs mainly via the αvβ3 integrin subtype present in these cells [11] as the therapeutic target of nanoformulations with RGD [41,45,55]. In addition, αvβ3-Integrin helps during the activation of MMPs, contributing to remodeling the ECM that facilitates endothelial cell migration [11]. Therefore, it is important, but not indispensable, for tumor angiogenesis [11], and blocking αvβ3 results in a drastic reduction in angiogenesis due to the inhibition of downstream signaling via the PAK/Src/Akt-pathway [100].

In GBM, overexpressed MMP helps tumor cells to survive, grow, and metastasize in distant sites [101], participating in the disruption and tumor neovascularization. Thus, the angiogenic response can be directly or indirectly mediated by MMPs through modulating the balance between pro- and anti-angiogenic factors [102]. Only the study by Séhédic [52] targeted a nanoformulation to MMP as an angiogenic marker, specifically MMP9.

The angiogenic process involves many signaling pathways, intracellular signaling pathways mediated by the growth factor of RTKS and intracellular proteins, as well as extra-cellular signaling pathways such as the notch/delta, ephrin/Eph receptor, roundabout/slit, and netrin/UNC (uncoordinated) receptor families, providing many opportunities for therapeutic intervention [103]. The study by Wang [55] developed a nanoformulation coupled with SWL that interfered in the EphA2 signaling pathway mediated angiogenesis process in GBM. This process occurs via the recruitment of phosphoinositide 3-kinase (PI3K) that stimulates downstream molecules of the Vav family of guanine nucleotide exchange factors (GEFs) and Rac1-GTP. Thus, the EphA2 signaling pathway promotes crosstalk with pro-angiogenic molecules such as growth factor receptors and adhesion molecules such as integrins and cadherins [104]. Other signaling pathways were also reported in the nanoformulation development [34,50,52,55,61,65], aiming to interfere in the regulation of tumor and angiogenic activities, mainly in signaling pathways involved with kinase enzymes. When activated by the phosphorylation process, they create a signal by transferring a phosphate group to a protein substrate. The study by Clavreul [34] used a multi-kinase inhibitor that interfered with the intracellular downstream serine/threonine kinases Raf-1, B-Raf, and mutant B-Raf through the application of SFN-loaded lipid nanocapsules, which intensively decreased the proportion of proliferating cells and tumor vessel area, and induced an early increase in tumor blood flow and vascular normalization, due to the correction of the structure and functionality of blood vessels, in which the increase in blood flow associated with tumor reduction prevents tumor cells from acquiring an aggressive phenotype by eliminating hypoxia, in addition to favoring a better effect of chemotherapeutic and radiotherapeutic agents [8]. Inhibition of the RAF1 gene by MiR-7-5p is associated with inhibition of vascular endothelial cell proliferation because Raf1 activates growth factor signaling downstream of the epidermal growth factor receptor (EGFR), a major drug target in GBM [105]. This miRNA is frequently downregulated in GBM microvasculature [106].

Microvessel tumor density and histopathological growth patterns during the angiogenic process can be analyzed using lectins such as WGA or the application of antibodies against specific antigens [9]. Most of the selected studies used the CD31 marker present in endothelial cells for this purpose [34,43,44,45,50,51,52,53,54,56,57,59,60,61,62,64,65,66], which can also detect antibodies against CD34 [55,58], von Willebrand factor [65], αvβ3-Integrin, or type IV collagen [9]. Detection of specific antigens ex vivo is relatively easy, either through chromogenic detection or fluorescent detection, but both methods have significant differences in their applicability. Through the IF, it is possible that several targets are detected simultaneously, with different fluorescence spectra, and targets are abundant or scarce. The study by Verreault [43] performed double detection of CD31 and Collagen IV after Irinophore C^TM^ administration and they quantified different structures involved in the angiogenic process: the extent of discontinuous basement membrane, the fraction of pericyte-uncovered blood vessels in tumor tissue, the blood vessel diameter, and the proportion of empty basement membrane sleeves that indicates the regression of pre-existing blood vessels suggesting a more homogenous distribution of blood across the entire tumor. It is also useful in epitope detection when the fluorescent liposomes used in nanoformulations are detected using the fluorochrome DiI applied to them, resulting in a higher dynamic range, as seen in Zhang’s work [45]. However, a clear disadvantage of IF compared to chromogenic detection of molecular targets is sensitivity. IHC benefits from signal amplification via indirect chromogenic detection, but it is not able to effectively mark multiple therapeutic targets simultaneously. As a result of this, the study by Séhédic [52] marked the CD68, NOSII, and Arg1 samples independently, differentiating TAMs of the antiangiogenic macrophages but without performing a quantification process. The study by Lin [57] detected several epitopes in isolation by IHC, providing more details on the action of CRLX101 (Cerulean Pharma, Cambridge, MA, USA) nanoformulation on the angiogenic process. In addition, the CD31 + endothelial cells were evaluated via the signaling cascade through the marking of CA IX, a promising endogenous hypoxia-related cell surface enzyme associated with increased necrosis [75,76]; this is considerably less common than ex vivo detection of the HIF1α marker, which was used in the study by Gao [41]. IHC was also the most frequent technique used for the detection of VEGF expelled by tumor cells [57,63,68], which was also detected by WB [57] and ELISA [32]. The study by Wu [50] showed a direct correlation between the VEGF protein level and the VEGF mRNA by transcriptomic analysis, which was also detected in the study by Kuang [58]. Transcriptomics can analyze different phases of the angiogenic process using qPCR, microarray, or RNA sequences [9]. By qPCR technique vascular signatures of genes originating from the tumor implantation and the stroma can be differentiated, due to a low overlap, with the help of bioinformatics analysis [9,107]. Blood perfusion evaluations were the focus when analyzing the antiangiogenic therapeutic process using different image techniques [34,43,48,53]. Results showed a decrease in proliferating cells and tumor vessel area after SFN-LNCs treatment [34], no blood flow existed in the tumor region following treatment with CARD-B6 [53], vessel dilation and hemorrhaging within the tumor exposed to VEGF-NSs and PEG-NSs [48], and restored basement membrane architecture (BMA) and reduced blood vessel diameter (BVD) of the tumor vasculature, suggesting restoration of the vessel architecture to a more normal state after Irinophore C^TM^ treatment, occurring vascular normalization [43].

Currently, a wide range of natural compounds has been recognized for the important role in the improvement in the efficacy of chemotherapeutics in combination therapy of GBM, highlighting the relative nontoxicity of the natural compounds when used in combination therapy, requiring lower dosage level, and showing as a potential solution for the multidrug resistance question that is a major cause of failure in cancer chemotherapies [108]. The PTX and DTX were the more representative drugs derived from plants, representing 36 and 18%, respectively of the selected studies that used drugs in the nanoformulations. These taxanes acting as anti-mitotic agents by promoting polymerization of microtubules and reducing depolymerization [109], as also as a radiation sensitized used for inhibiting the GBM growth and proliferation when associated with radiation therapy. In addition, the plant-based medicines used in angiotherapy, mainly GBM therapy reported in the selected studies already are well described in the literature, regarding its antiangiogenic and antimetastatic action that occurs targeting different molecular pathways, for example, the inhibiting VEGF production and the expression of HIF-1by taxanes, inhibition of the action of Topoisomerase I by alkaloids, suppressing the production of plasminogen activator (PA) and PA inhibitor-1 by flavonoids, among other targets of angiogenesis pathways [110,111,112]. The selected studies that used these natural drugs associated with nanoformulations [38,41,43,50,53,54,56,57,59,61,62], showed that the nanocarriers overcame the biochemical/biophysical barriers, allowing a supply of medicines throughout the tumor region through the BBB. The properties of the nanocarriers can be modified for selective and controlled release of drugs with minor effects [113]. The search for better responses in the treatment of GBM makes these studies that associate natural drugs and nanoformulations extremely promising.

Analysis of the studies included in this review showed that the applications of nanoformulations with drug delivery strategies aimed at interfering in the angiogenic process in pre-clinical GBM models were very diverse in almost all aspects addressed, making it difficult to define the best antiangiogenic strategy. The tumor models of the studies were induced with different cell lines and subjected to various chemotherapeutic agents coupled with nanoformulations designed to achieve different therapeutic targets and exert equally diverse mechanisms. On the other hand, in relation to angiogenesis, there is little variety in the methods of analyzing the therapeutic effects of nanoformulations on the various processes involved in angiogenesis, both in global and specific ways such as the process of hypoxia, recruitment of inflammatory cells, angiogenic growth factor production, basement membrane degradation, endothelial cell migration, and proliferation, differentiation, and modulation of vascular support cells. Therefore, the present review highlighted the need for standardization between different antiangiogenic therapy approaches for the GBM model. This would allow for more effective analyses, especially meta-analyses, and it would help in future translational studies. Currently, only the biodegradable polymeric nanoformulation of carmustine wafers is used in clinical practice into the ressection cavity, presenting controversial results until recently, although meta-analyzes have shown increased patient survival compared to control groups, increasing the effectiveness of traditional therapy [114].

## 4. Materials and Methods

### 4.1. Search Strategy

This systematic review followed the Preferred Reporting Items for Systematic Reviews and Meta-Analyses (PRISMA) Guideline [49]. We performed a search for indexed articles published prior to February 2020 in the databases: PubMed, Scopus, and Cochrane using the following search criteria: (Glioblastoma OR Glioma) AND (Therapy OR Therapeutic) AND (Nanoparticle) AND (Antiangiogenic OR Angiogenesis OR Anti-angiogenic). Then, we applied the following boolean operators (DecS/MeSH) and keywords sequence in the search of each database:

PubMed: (((((((((angiogenesis) OR antiangiogenesis) OR angiogenic) OR antiangiogenic) OR anti-angiogenic) OR anti-angiogenesis)) AND ((((glioblastoma) OR glioma)) OR gliomas)) AND ((nanoparticle) OR nanoparticles)) AND ((((((((((((therapy) OR therapeutic) OR therapeutics) OR therapies) OR theranostic) OR theranostics) OR nanotherapeutic) OR nanotherapy) OR nanotherapeutics) OR nanotherapies)) OR treatment).

SCOPUS: ((TITLE-ABS-KEY (glioblastoma) OR TITLE-ABS-KEY (glioma) OR TITLE-ABS-KEY (gliomas))) AND ((TITLE-ABS-KEY (nanoparticle) OR TITLE-ABS-KEY (nanoparticles))) AND ((TITLE-ABS-KEY (angiogenesis) OR TITLE-ABS-KEY (angiogenic) OR TITLE-ABS-KEY (antiangiogenic) OR TITLE-ABS-KEY (antiangiogenesis) OR TITLE-ABS-KEY (anti-angiogenic ) OR TITLE-ABS-KEY (anti-angiogenesis))) AND ((TITLE-ABS-KEY (therapy) OR TITLE-ABS-KEY (therapeutic) OR TITLE-ABS-KEY (therapeutics) OR TITLE-ABS-KEY (therapies) OR TITLE-ABS-KEY (theranostic) OR TITLE-ABS-KEY (theranostics) OR TITLE-ABS-KEY (nanotherapeutic) OR TITLE-ABS-KEY (nanotherapy) OR TITLE-ABS-KEY (nanotherapeutics) OR TITLE-ABS-KEY (nanotherapies))).

Cochrane: “glioma” in Title Abstract Keyword OR “glioblastoma” in Title Abstract Keyword AND “angiogenesis” in Title Abstract Keyword AND “therapy” in Title Abstract Keyword AND “nanoparticle” in Title Abstract Keyword (Word variations have been searched).

### 4.2. Inclusion Criteria

This review included only original articles written in English that had used: (i) antiangiogenic therapy using nanoparticles, (ii) detection of the effect of therapy using angiogenic markers, (iii) therapies for glioma or GBM, and (iv) in vivo studies with intracranial induction of glioma or GBM.

### 4.3. Exclusion Criteria

Reasons for excluding studies were as follows: (i) nanodiagnosis, (ii) review articles, (iii) book chapters, (iv) protocols, (v) editorials/expert opinions/notes, (vi) letters/communications/conference abstracts, (vii) publications in languages other than English, (viii) studies that did not analyze angiogenesis, (ix) studies with tumor induction in the flank/subcutaneous, (x) in vitro studies only, (xi) use of tumors other than glioma or GBM, and (xii) non-angiogenic targets.

### 4.4. Data Extraction, Data Collection, and Risk of Bias Assessment

In this systematic review, four authors (G.N.A.R.; A.H.A.; M.P.N.; and J.B.M.), selected in pairs, independently and randomly, reviewed and evaluated the titles and abstracts of the publications identified by the search strategy in the databases mentioned above and all potentially relevant publications were retrieved in full. These same authors evaluated the full-text articles to decide whether the eligibility criteria were met. Discrepancies in study selection and data extraction between the four authors were discussed with two other authors (F.A.O. and L.F.G.) and resolved.

G.N.A.R., A.H.A., and F.A.O. analyzed studies on tumor cells and glioma models in vivo; J.B.M. and M.P.N. analyzed studies on the use of nanoparticles for anti-angiogenic therapy and the antiangiogenic therapy effect, and G.N.A.R., A.H.A., and M.P.N. analyzed angiogenic effects evaluation. The analysis of the articles and the preparation of the tables were carried out by consensus. In each case of disagreement, a fifth independent and senior author (L.F.G.) decided by data addition or subtraction. The final inclusion of studies in this review was in agreement with all authors.

### 4.5. Data Analysis

All results were described and presented using the percentage distribution for all variables analyzed in the tables.

## Figures and Tables

**Figure 1 ijms-21-04490-f001:**
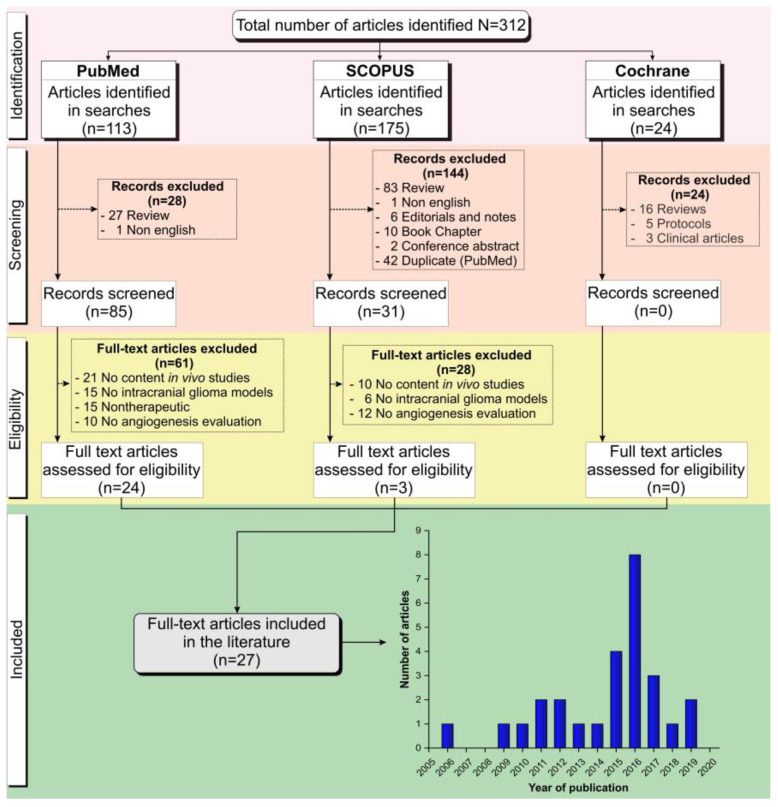
Flowchart corresponding to the stages of the Preferred Reporting Items for Systematic Reviews and Meta-Analyses (PRISMA) guidelines [49] of the article screening process for inclusion in this review.

**Figure 2 ijms-21-04490-f002:**
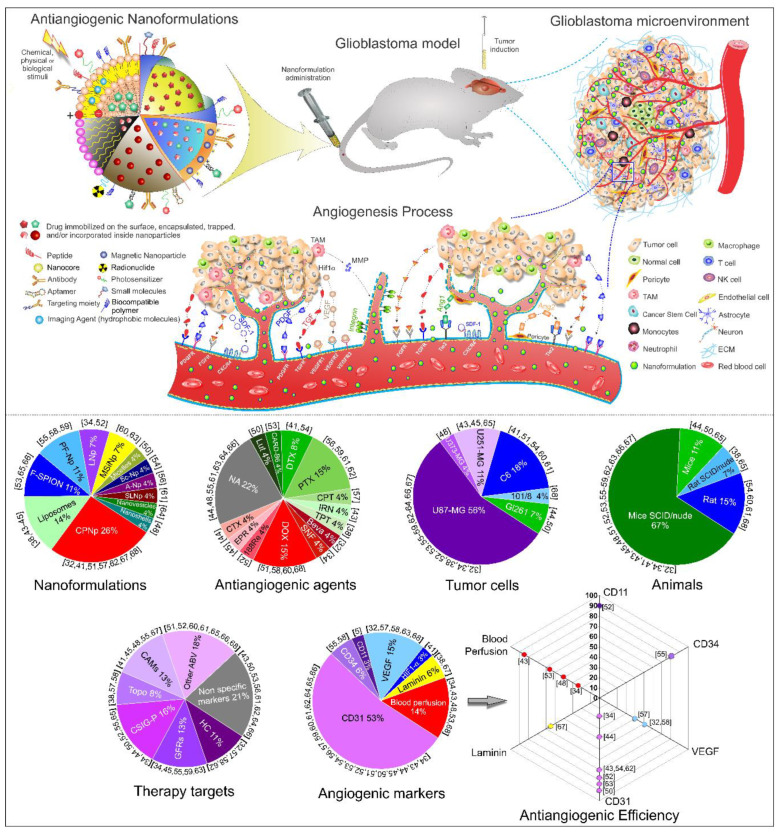
Schematic representation of antiangiogenic nanoformulations for glioblastoma therapy from a pre-clinical approach and the angiogenic process in the tumor microenvironment and the pie charts of the main results found in the systematic review, as well as the quantification of antiangiogenic efficiency appointed by each study. Abbreviations—188Re: Rhenium-188; ABV: Angiogenic blood vessels; Ang1: Angiopoietin 1; Ang2: Angiopoietin 2; A-Np: Functionalized albumin nanoparticles; Beva: Bevacizumab; CAMs: Cell adhesion molecules; CARD: Nanoparticles with B6 loading three drugs ((CA4+AZO-ATRA+DOX+SPIONs)NPs); CPNp: Complex Polymeric Nanoparticles; CPT: Camptothecin; CSIG-P: Cell signaling pathways; CTX: Chlorotoxin; CXCR4: C-X-C chemokine receptor type 4 (CD184); DOX: Doxorubicin; DTX: Docetaxel; ECM: Extracellular matrix; EPR: Epirubicin; FGF: Fibroblast growth factor; FGFR: FGF receptor; F-SPIONs: Functionalized SPIONs; GFRs: Growth factor receptors; HC: Hypoxic cascade; HIF1α: Hypoxia -inducible factor 1 α; IRN: Irinotecan; LNp: Lipid Nanocapsules; Lut: Luteolin; MMP: Metaloproteinase matrix; MSiNp: Multifunctional sílica based nanoparticles; NK cell: Natural killer cell; PDGF: Platelet-derived growth factor; PDGFR: PDGF Receptor; PF-Np: Peptide functional nanoparticles; PTX: Paclitaxel; SCID: Severe Combined Immunodeficiency; Sc-Np: Scalffold Nanoparticles; SDF-1: Stromal cell-derived factor 1; SLNp: Solid lipid nanoparticles; SNF: Sorafenib; TAM: Tumor-associated macrophage; TGF: Transforming growth factor; TGFR: TGF receptor; Tie 2: Ang1 and Ang2 receptors; Topo: Topoisomerase; TPT: Topotecan; VEGF: Vascular endothelial growth factor; VEGFR1: VEGF Receptor type 1; VEGFR2: VEGF Receptor type 2; VEGFR3: VEGF Receptor type 3.

**Table 1 ijms-21-04490-t001:** Cell characteristics of glioblastoma induction.

Ref.	Year	Tumor Cell	Source of Cells	Medium Culture	Supplement	Cell Modifications
Wu et al. [50]	2019	GL261	Mouse	DMEM	10%FBS	NA
Sousa et al. [32]	2019	U87-MG	Human	DMEM	10%FBS; 10 μg/mL Blast	Luciferase expression
Clavreul et al. [34]	2018	U87-MG	Human	DMEM-HG	10%FBS; 1% antibiotics	NA
Sun et al. [51]	2017	C6	Rattus norvegicus	DMEM	10%FBS; 2 mM Glut; 100 U/mL Pen; 100 mg/mL Strep	NA
Séhédic et al. [52]	2017	U87-MG	Human	DMEM-HG	10%FBS; l-Glut; 10 U/mL Pen; 10 mg/mL Strep; 25 μg/mL Amp-B	Expression of CXCR4 and RFP
Lu et al. [53]	2017	U87-MG	Human	DMEM	10%FBS; 1% Pen/Strep	NA
Xu et al. [54]	2016	C6	Rattus norvegicus	DMEM-HG	FBS	NA
Wang et al. [55]	2016	U87-MG	Human	DMEM	10%FBS	Luciferase expression
Lin et al. [56]	2016	U87-MG	Human	DMEM	10%FBS; 1% antibiotics	Luciferase expression
Lin et al. [57]	2016	U87-MG	Human	MEM	10%FBS; 2 mM l-Glut; 100 U/mL Pen; 100 mg/mL Strep; 1 mM SP; 1 mM NAA	Luciferase expression
Kuang et al. [58]	2016	U87-MG	Human	NR	NR	Luciferase expression
Hu et al. [59]	2016	U87-MG	Human	DMEM	FBS; Pen/Strep	NA
Hu et al. [60]	2016	C6	Rattus norvegicus	NR	NR	NA
Banerjee et al. [61]	2016	C6	Rattus norvegicus	DMEM	10%FBS; 1 mM Glut; 100 U/mL Pen; 100 ng/mL Strep	NA
Zhang et al. [45]	2015	U251-MG	Human	MEM-EBSS	10%FBS	NA
Feng et al. [62]	2015	U87-MG	Human	DMEM-HG	10%FBS; 100 U/mL Pen; 100 µg/mL Strep	NA
Costa et al. [44]	2015	GL261	Mouse	DMEM-HG	10%FBS; 100 U/mL Pen; 100 µg/mL Strep; 100 mM HEPES; 12 mM NaHCO_3_	NA
Bechet et al. [63]	2015	U87-MG	Human	NR	NR	NA
Gao et al. [41]	2014	C6	Rattus norvegicus	DMEM-HG	FBS	Expression RFP
Wojton et al. [64]	2013	U87-MG	Human	DMEM	10%FBS; 100 U/mL Pen; 10 mg/mL Strep	U87ΔEGFR; U87ΔEGFR-Luc
Janic et al. [65]	2012	U251-MG	Human	DMEM	10%FBS	NA
Day et al. [48]	2012	U373-MG	Human	RPMI 1640	10%FBS; 1% Glut	Luciferase expression
Verreault et al. [43]	2011	U251-MG	Human	DMEM	10%FBS; 1% l-Glut; 1% Pen; 1% Strep	NA
Agemy et al. [66]	2011	U87-MG	Human	DMEM-F12	10%FBS; 1% Glut; 1% Pen; 1% Strep	Luciferase expression
		005	Mouse	DMEM-F12	1%N_2_; 20 ng/mL FGF-2; 20 ng/mL EGF; 40 μg/mL heparin	Luciferase expression
		Spheres#	Human	DMEM-F12	l-Glut; 0.3% Gluc; 50 μg/mL Pen/Strep; 0.1 mg/mL Apo-transf; 20 nM Prog; 30 nM Na_2_SeO_3_, 60 μM Put; 25 μM/mL Ins; 3 mM NaHCO_3_; 10 mM HEPES; 20 ng/mL EGF, 10 ng/mL LIF; 20 ng/mL FGF	NA
		H-RasV12-shp53 lentivirus	NA	NA	NA	Luciferase expression
Ding et al. [67]	2010	U87-MG	Human	MEM	10% FBS	NA
Hekmatara et al. [68]	2009	101/8*	Rattus norvegicus	NA	NA	NA
Saito et al. [38]	2006	U87-MG	Human	MEM	10% FBS; 100 U/mL; 0.1 mg/mL Strep	NA

Abbreviations—Ref.: reference; NA: not applicable; NR: not reported; DMEM: Dulbecco’s modified Eagle’s medium; DMEM-HG: DMEM-high glucose; MEM: minimum essential medium; MEM-EBSS: Eagle’s minimum essential medium with Earle’s balanced salts; RPMI 1640: Medium Roswell Park Memorial Institute 1640; DMEM-F12: Dulbecco’s Modified Eagle Medium/Nutrient Mixture F-12; FBS: Fetal Bovine Serum; Blast: Blasticidin; Glut: Glutamine; Pen: Penicillin; Strep: Streptomycin; l-Glut: l-Glutamine; Amp-B: Amphotericin-B; SP: Sodium pyruvate; NAA: Nonessential Amino Acids; HEPES: (4-(2-hydroxyethyl)-1-piperazineethanesulfonic acid); NaHCO_3_: Sodium bicarbonate; EGF: Epidermal Growth Factor; LIF: Leukemia Inhibitory Factor; FGF: Fibroblast Growth Factors; N2: N2 supplement; Gluc: Glucose; Apo-transf: Apo-transferrin; Prog: progesterone; Na_2_SeO_3_: Sodium Selenite; Put: putrescine; Ins: Insulin; CXCR4: C-X-C chemokine receptor type 4; RFP: Red Fluorescent Protein; U87ΔEGFR: Normal expression of the epidermal growth factor receptor; U87ΔEGFR-Luc: Normal expression of the epidermal growth factor receptor marked luciferase. Note—*The rat 101/8 glioblastoma is an orthotopic model initially generated by the injection of α-dimethylbenzanthracene (α-DMBA) into the cerebellum of Wistar rats followed by serial transplantation of tumor tissues into the hemisphere of Wistar rats; #Human GBM spheres lines BT37, BT70, and BT7.

**Table 2 ijms-21-04490-t002:** Glioblastoma induction model characteristics.

Ref.	Animal Description						Glioblastoma Induction						
	Animal	Specie	Sex	Weight (g)	Age (week)	n/N	Cell Type	Cell Number (cell/mL)	AV (μL)	Vehicle	AT (min)	Local Administration	Coordinates (AP; ML; DV:mm)
Wu et al. [50]	Mice	C57BL/6	F	NR	6–8	7/35	GL261	2 × 10^7^	5	DMEM	NR	NR	NR
Sousa et al. [32]	Mice	Nude athymic	NR	26–32	6–8	3–4/12–16	U87-MG	5 × 10^5^	5	NR	NR	L. Cerebral hemisphere	1.5; 2; 3.5
Clavreul et al. [34]	Mice	Swiss nude	F	22–23	8–10	5–7/22	U87-MG	5 × 10^4^	5	HBSS+	NR	R. Striatum	0.5; 2.1; 3
Sun et al. [51]	Mice	Nude athymic	NR	NR	NR	7/28	C6	1 × 10^4^	5	PBS	NR	R. Striatum	NR
Séhédic et al. [52]	Mice	CB17-SCID	F	NR	8	6–9/47	U87-MG	5 × 10^4^	5	EMEM	NR	R. Striatum	0.5; 2; 3
Lu et al. [53]	Mice	BALB/C nude	F	22–23	NR	NR/NR	U87-MG	5 × 10^5^	5	MEM	NR	NR	3; 3; 3
Xu et al. [54]	Rat	Sprague-Dawley	M	250–350	NR	12/60	C6	1 × 10^6^	10	NR	10	NR	1; 3; 5
Wang et al. [55]	Mice	Nude athymic	NR	NR	4–6	11/30	U87-MG	2.4×10^5^	8	PBS	>1	NR	1; 2; 3
Lin et al. [56]	Mice	BALB/C nude	NR	NR	3–4	4/40	U87-MG	NR	NR	NR	NR	L. Cerebral hemisphere	NR
Lin et al. [57]	Mice	BALB/C nude	F	NR	6	5/25	U87-MG	2 × 10^5^	3	PBS	NR	R. Frontal lobe	1; 2; 3
Kuang et al. [58]	Mice	BALB/C nude	M	20–25	NR	15/105	U87-MG	NR	NR	DMEM	NR	NR	NR
Hu et al. [59]	Mice	BALB/C nude	M	NR	4	3/12	U87-MG	5 × 10^5^	5	PBS	NR	R. Striatum	NR; 1.8; 3
Hu et al. [60]	Rat	Sprague-Dawley	M	NR	4	16/64	C6	1 × 10^6^	10	PBS	10	R. Cerebral hemisphere	NR; 2; 5
Banerjee et al. [61]	Rat	Sprague-Dawley	F	200–220	27	9/36	C6	1 × 10^6^	5	NR	NR	R. Cerebral hemisphere	2; 2; 3
Zhang et al. [45]	Mice	BALB/C nude	M	18–20	NR	3/15	U251-MG	6 × 10^5^	3	MEM-EBSS	3	R. Cerebral hemisphere	1.5; 1.8; 3
Feng et al. [62]	Mice	BALB/C nude	M	18–22	NR	6/20	U87-MG	5 × 10^5^	5	PBS	NR	R. Striatum	NR
Costa et al. [44]	Mice	C57BL/6	M	NR	8	6–8/NR	GL261	1.25 × 10^5^	3	NR	15	R. Cerebral hemisphere	−1.06; 3; 3
Bechet et al. [63]	Mice	Nude athymic	M	150–180	8	NR/NR	U87-MG	5 × 10^4^	5	HBSS+	25	Parenchyma	0.5; 2.7; 4.4
Gao et al. [41]	Mice	BALB/C nude	M	18–22	4–5	13/78	C6	5 × 10^5^	5	NR	3	R. Striatum	NR
Wojton et al. [64]	Mice	Mut6¥ /cKO	F; M	NR	NR	3–4/NR	U87-MG	1 × 10^5^	NR	NR	NR	NR	NR; 2; 3
Janic et al. [65]	Rat	Nude athymic*	NR	150–170	6–8	10/NR	U251-MG	4 × 10^5^	5	DMEM	5	R. Cerebral hemisphere	1; 3; 2.5–3.5
Day et al. [48]	Mice	ICR-PrkdcSCID	M	NR	NR	3/9	U373-MG	1 × 10^5^	NR	RPMI-1640	NR	NR	2; 1; 3
Verreault et al. [43]	Mice	Rag2M	F	NR	7–10	5–6/9	U251-MG	7.5 × 10^4^	NR	NR	NR	R. Caudate nucleus-putamen	1; −1.5; −3.5
Agemy et al. [66]	Mice	NOD-SCID	NR	NR	NR	8–10/16–20	U87-MG	NR	NR	NR	NR	R. Hippocampus	NR
	Mice	NOD-SCID	NR	NR	NR	8–10/16–20	005	3 × 10^5^	1.5	PBS	NR	R. Hippocampus	NR
	Mice	NOD-SCID	NR	NR	NR	8–10/16–20	Spheres#	5 × 10^5^	1.5	PBS	NR	R. Hippocampus	NR
	Mice	NOD-SCID	NR	NR	NR	8–10/16–20	H-RasV12-shp53 lentivirus £	NA	NR	NR	NR	R. Hippocampus	NR
Ding et al. [67]	Mice	Nude Athymic$	NR	NR	NR	8/24	U87-MG	5 × 10^4^	NR	NR	NR	R. Basal ganglia field	NR
Hekmatara et al. [68]	Rat	Wistar	M	200–250	NR	18–20/58	101/8	1 × 10^6^	NR	NR	NR	R. Lateral ventricle	2; 2; 4
Saito et al. [38]	Rat	Nude athymic*	M	250	NR	7/35	U87-MG	5 × 10^5^	2 × 5	HBSS-	2 × 2	Striatum	0.5; 3; 4–4.5

Abbreviations—Ref.: Reference; n/N: number of animals per group/total number of animals; Av: Administration Volume; AT: Administration Time; AP: Anteroposterior; ML: Mediolateral; DV: Dorsoventral; NA: Not Applicable; NR: Not Related; F: Female; M: Male; DMEM: Dulbecco’s Modified Eagle’s Medium; HBSS+: Hank’s Balanced Salt Solution with Ca^2+^ and Mg^2+^; PBS: Phosphate-Buffered Salin; EMEM: Eagle’s Minimum Essential Medium; MEM: Minimum Essential Medium; MEM-EBSS: Powdered GE Healthcare HyClone Minimal Essential Medium with Earle’s; RPMI-1640: Medium Roswell Park Memorial Institute 1640; HBSS-: Hank’s Balanced Salt Solution without Ca^2+^ and Mg^2+^; L: Left Cerebral Hemisphere; R: Right Cerebral Hemisphere. Note—^¥^Pten^loxP/loxP^ females to generate Mut6 mice (GFAP-cre; Nf1^loxP/+^; Trp53^−/loxP^; Pten^loxP/+^) and quadruple cKO mice GFAP-CreER; Pten^loxP/loxP^; Trp53^loxP/loxP^; Rb1^loxP/loxP^ mice were bred with Rb1^-/-^ mice (p107-null); *Nude athymic (rnu-/rnu-); ^$^Nude Athymic CrTac:NCr-Foxn1nu homozygous; #Human GBM spheres lines BT37, BT70, and BT7; ^£^lentiviral vector expressing H-RasV12 oncogene and an shRNA targeting p53 (H-RasV12-shp53).

**Table 3 ijms-21-04490-t003:** Nanoformulations used in the antiangiogenesis therapy.

Ref.	Particle	Drug	Formulations	Manufacture	Size (nm)	ζ (mV)	PDI	EE (%)	DLE (%)	Release of the Drugs
Wu et al. [50]	Micelles	Luteolin	**(Lut/Fa-PEG-PCL)**	Synthesized	34.7	−9.2	0.12	98.5	5	Luteolin: 46%(PBS 0.5% Tween-80, 10 h, 37 °C)
Sousa et al. [32]	Polymeric Nanoparticles	Bevacizumab	**(Beva-loaded PLGA)**	Synthesized	185	−1.6	0.056	82.47	1.62	Bevacizumab: 14%(after 7 days in vitro study, pH 7.4)
Clavreul et al. [34]	Lipid nanocapsules	Sorafenib	**(SFN-LNCs)**	Synthesized	54	−7.8	0.15	105	NR	SFN: 11%(DPBS, 8 h) and 20% (DPBS, 120 h)
Sun et al. [51]	Polymeric Nanoparticles	DOX	DOX-NP		110	−29.7		56.33	1.43	DOX:~70%(PBS, pH7.4, 72 h, 37 °C) and ~80%(10% rat plasma, 72 h, 37 °C)
			**(AP1-DOX-NP)**		120	−26.3		53.74	1.37	
Séhédic et al. [52]	Lipid nanocapsules Nanocarriers	^188^Re	LNC	Synthesized	55.41	−4.51	0.03	NA	NA	**Antibodies per LNC:**
			12G5-LNC		60.44	−13.87	0.24			**35%**
			IgG2a-LNC		63.48	−14.95	0.26			**15%**
			LNC^188^Re		58.12	−8.37	0.05			**-**
			**(12G5-LNC^188^Re)**		77.25	−24.77	0.21			**13%**
			IgG2a-LNC^188^Re		74.81	−26.23	0.21			**10%**
Lu et al. [53]	Functionalized SPIONs	ATRA; CA4; DOX	**(CARD** **-** **B6)**	Synthesized	<100	−0.5–0.9	NR	CA4: 75.3 ATRA: 77.8 DOX: 78.4	PAD: 58.37 PAAA: 14.53 PABA: 16.23	CA4:64.47%(pH 6.5, 12 h, 37°C DOX:68.37%(pH5.0, 48 h, 37 °C) ATRA: 85.11%(hypoxic condition, pH 5.0, 12 h, 37 °C) DOX: 66.39%(hypoxic condition, pH 5.0, 48 h, 37 °C)
Xu et al. [54]	DTX-NPs-adsorbing dBECM scaffold	DTX	**(DTX-NPs)**	Synthesized	32.0	17.7	NR	73.37	3.99	DTX: 38%(pH 7.4, 24 h, 37 °C)
Wang et al. [55]	Multi-functional nanoparticles	NA	**H-S-R NPs1**	Synthesized	164	−31.77	0.119	NA	NA	NA
			H-S-R NPs2		190	−29	0.128			
Lin et al. [56]	LMWP-modified albumin nanoparticles	PTX; 4-HPR	BSA NPs	Synthesized	140	−30	0.089	51.63	3.6	PTX: 70%(PBS, pH 7.4, 0.5% *w*/*v* SDS, 96 h, 37 °C)
			**(L-BSA NPs)**		145	−16	0.074	53.24	3.7	PTX: 73%(PBS, pH 7.4, 0.5% *w*/*v* SDS, 96 h, 37 °C)
Lin et al. [57]	Polymeric Nanoparticles	CPT	**(CRLX101)**	Cerulean Pharma, Cambridge, MA	20–30	−6	NR	NR	10	NR
Kuang et al. [58]	Dual Functional peptide-driven nanoparticles	DOX; shVEGF	**(DGL-PEG-T7/shVEGF-DOX)**	Synthesized	142.9	NR	NR	NR	NR	NR
Hu et al. [59]	Peptide dual-decorated nanoparticle	PTX	NP		102	−37.5	0.13	NA	NA	PTX: 78.3%(1 mL of PBS/39 mL of release medium, 72 h)
			ATWLPPR-NP		113	−31.6	0.14			PTX: 77.1%(1 mL of PBS/39 mL of release medium, 72 h)
			CGKRK-NP		119	−14.6	0.17			PTX: 79.4%(1 mL of PBS/39 mL of release medium, 72 h)
			**(AC-NP)**		123	−11.4	0.15			PTX: 75.6%(1 mL of PBS/39 mL of release medium, 72 h)
Hu et al. [60]	Functionalized mesoporous nanoparticles	DOX	MSN-DOX-PDA	Synthesized	156.1	−24		44.84	19.02	DOX:50%(Acetate, pH 4.5, 24 h, 37 °C)
			**(MSN-DOX-PDA-NGR)**		168	−22				
Banerjee et al. [61]	Solid lipid Nanoparticles	PTX	PSLN	Synthesized	158	−24.8	0.16	88	5.18	NR
			**(PSM)**		178	−17.4	0.19	86	5.06	
Zhang et al. [45]	Liposomes	Epirubicin	Epirubicin liposomes	Synthesized	97.92	−14.3	0.24	96.88	NR	Epirubicin:<1%(PBS 10% FBS, Ph 7.4,2 h, 37 °C) ~2%(PBS 10% FBS, Ph 7.4, 36 h, 37 °C)
			Glu-targeting epirubicin liposomes		108.87	−14.6	0.20	97.81		
			cRGD-targeting epirubicin liposomes		110.91	−9.63	0.21	98.30		
			**(Functional targeting epirubicin liposomes)**		108.97	−15.2	0.23	98.68		
Feng et al. [62]	Polymeric Nanoparticles	PTX	NP-PTX	Synthesized	109.76	−33.35	0.092	47.07	1.49	PTX: 69.25%(PBS 0.5% Tween 80, Ph 7.4, 96 h, 37 °C) and 82.91%(10% rat plasma, 96 h, 37 °C)
			**(CooP-NP-PTX)**		118.95	−27.59	0.157	44	1.31	PTX: 73.52%(PBS 0.5% Tween 80, pH7.4, 96 h, 37 °C) and 84.53% (10% rat plasma, 96 h, 37 °C)
Costa et al. [44]	Liposomes	NA	**(CTX-coupled SNALPs)**	Synthesized	<190	NR	0.3	85-95	NA	NA
Bechet et al. [63]	Multifunctional Silica-based nanoparticles	Chlorin (photosensitizer)	NP-TPC	Synthesized	2.9	42.9	NR	NR	NR	NR
			**(NP-TPC-ATWLPPR)**			22.6				
Gao et al. [41]	Polymeric nanoparticles		Coumarin-6-NPs	Synthesized	101.3	−9.07	0.191	NA	NA	NA
			Coumarin-6-ILNPs		105.6	−10.12	0.201			
			Coumarin-6-IRNPs		112.4	−9.86	0.224			
			Coumarin-6-RNPs		111.8	−11.21	0.199			
		DTX	DTX-NPs		120.1	−8.77	0.173			
			DTX-ILNPs		131.2	−8.89	0.181			
			**(DTX-IRNPs)**		137.9	−9.76	0.192			
			DTX-RNPs		127.6	−7.81	0.187			
Wojton et al. [64]	Nanovesicles	NA	**(SapC-DOPS nanovesicles)**	Synthesized	NR	NR	NR	NA	NA	NA
Janic et al. [65]	SPIONs	NA	**(FePro)**	Feridex IV; Bayer-Schering Pharma, Wayne, NJ, USA	141.8	−31.30	0.285	NA	NA	NA
Day et al. [48]	Nanoshells	NA	Bare NS	Synthesized	162.4	−57.9	NR	NA	NA	NA
			**(VEGF-NS)**		188.0	−33.4				
			PEG-NS		196.8	−32.7				
Verreault et al. [43]	Liposomes	Irinotecan, DOX and vincristine	**(Liposomal Irinophore C^TM^,** Caelyx^®^ and Vincristine)	Caelyx^®^, Schering-Plough, QC, Canada	NR	NR	NR	NR	NR	NR
Agemy et al. [66]	Functionalized SPIONs	Mitochondria-targeted D[KLAKLAK]_2_ peptide	**(Iron Oxide Nanoworms-** **CGKRKD(KLAKLAK)_2_)**	Synthesized	NR	NR	NR	NA	NA	NA
Ding et al. [67]	Polymeric Nanoparticles	NA	PMLA-→ (**P/LLL/AON/Hu/Ms, P/LOEt/AON/Hu/Ms)**	Synthesized	6.6→18 22	−27→ −9.4–5.2	NR	NA	NA	NA
Hekmatara et al. [68]	Polymeric Nanoparticles	DOX	**(DOX-np)**	Synthesized	260	−19	0.02	70	NR	NR
Saito et al. [38]	Liposomes	TPT	**(Ls-TPT)**	Hermes Bioscience, Inc. (South San Francisco, Calif.)	NR	NR	NR	>95	NR	NR

Abbreviations—Ref.: Reference; ζ: zeta potential; PDI: Polydispersity index; EE: Encapsulation Efficiency; DLE: Drug Loading Efficiency; NR: Not Related; NA: not applicable; Fa-PEG-PCL: folic acid modified poly(ethylene glycol)-poly(e-caprolactone); Lut: luteolin; PBS: Phosphate Buffered Saline; Beva: Bevacizumab; PLGA: Poly(d,l-lactic-co-glycolic) acid; SFN: Sorafenib; LNCs: lipid nanocapsules; DPBS: Dulbecco’s Phosphate Buffered Saline; DOX: Doxorubicin; NP: Nanoparticles; AP1: CRKRLDRNC peptide; LNC: Lipid nanocapsules; 12G5: (CD184, #555971, BD Pharmingen); IgG2a: (BD Biosciences, Le Pont-de-Claix, France); 188Re: Rhenium-188; B6: peptide motif B6 (1,2-dioleoylsn-glycero-3-phosphoethan-olamine-n-[poly (ethylene glycol)] 2000 (DSPE-PEG2000)); CARD: nanoparticles with B6 loading three drugs ((CA4+AZO-ATRA+DOX+SPIONs)NPs); CA4: Combretastatin A4; AZO-ATRA: Azobenzene - ll-trans retinoic acid; PAD:PAE-A-DOX; PAAA:PAE-A-AZO-ATRA; PABA:PAE-A-BZD-ATRA; PAE: poly (β-amino ester); A: amido bond; BZD: Benzidine; DTX: docetaxel; DTX-NPs: DTX-loaded nanoparticles; H-S-R Nps: heparin-containing NPs with two ligands (SWL and cRGD); cRGD: cyclic arginine-glycine-aspartate motif; PTX: Paclitaxel; 4-HPR: fenretinide; BSA: Bovine serum albumin; L-BSA: LMWP(sequence: CVSRRRRRRGGRRRR)–BSA; CPT: Camptothecin; CRLX101: nanoparticle-drug conjugate (NDC), containing approximately 10 wt% CPT conjugated to a linear, cyclodextrin-polyethylene glycol (CD-PEG) copolymer; shVEGF: Plasmid shVEGF; DGL: dendrigraft poly-l-lysines; PEG: polyethylene glycol; T7: peptide T7 (sequence His-Ala-Lle-Tyr-Pro-Arg-His); AC-NP: ATWLPPR and CGKRK peptide dual-decorated nanoparticulate drug delivery system; ATWLPPR: H-Ala-Thr-Trp-Leu-Pro-Pro-Arg-OH; CGKRK: (Cys-Gly-Lys-Arg-Lys) peptide; NGR: peptide (CYGGRGNG); PDA: polydopamine; MSN: Mesoporous Silica Nanoparticles; PSM: PTX-loaded solid lipid nanoparticles (SLN) modified with Tyr-3-octreotide (TOC); PSLN: PTX loaded SLN; Glu: glucose (4-aminophenyl β-d-glucopyranoside); FBS: Fetal Bovine Serum; CooP: peptide (ACGLSGLGVA)X; CTX-coupled SNALPs: chlorotoxin (CTX)-coupled (targeted) stable nucleic acid lipid particle (SNALP); NP-TPC: Nanoparticle- 5-(4-carboxyphenyl)-10,15,20-triphenylchlorin; ILNPs: IL-13p conjugated PEG-PCL nanoparticles; RNPs: RGD conjugated PEG-PCL nanoparticles; IRNPs: IL-13p and RGD conjugated PEG-PCL nanoparticles; SapC-DOPS: Saposin C-dioleoylphosphatidylserine; FePro: Ferumoxides-Protamine Sulfate; NS: Nanoshells (silica core/gold shell nanoparticles); VEGF: Vascular endothelial growth factor; D[KLAKLAK]2: The α-helical amphipathic peptide D[KLAKLAK]2; CGKRK: The CGKRK (Cys-Gly-Lys-Arg-Lys) peptide; P: PMLA (poly(β-l-malic acid)); LLL: H2N-Leu-Leu-Leu-OH; LOEt: H2N-Leu-Leu-Leu-NH2, H2N-Leu-ethylester; AON: antisense oligonucleotide; Ms: mAb (Ms) targeting blood–brain tumor barrier endothelium (mouse TfR); Hu: mAb (Hu) targeting tumor cells (human TfR); TfR: transferrin receptor; Dox-np: Doxorubicin-loaded polysorbate 80-coated poly(butyl cyanoacrylate) (PBCA) nanoparticles; Ls-TPT: liposomal topotecan. Note: The formulations highlighted in bold were the ones that had greater efficiency.

**Table 4 ijms-21-04490-t004:** Antiangiogenic therapeutic process for glioblastoma.

Ref.	Therapy Type (NP: Formulations)	Therapeutic Target	Route/Local of Administration	Frequency- Dose (mg/Kg)	Vehicle	Time Point of Therapy	Tumoral Reduction	Follow-Up Evaluation after Induction	Therapeutic Evaluation Techniques
Wu et al. [50]	Drug delivery (Lut/Fa-PEG-PCL)	Signal transduction pathways thar regulates tumor activities	Tail vein	Daily 50	Saline	5th to 13th day	NR	5th to 13th day (each 2 days) -Until cachexia of the mouse appeared	FLI; TUNNEL assay; Survival curve
Sousa et al. [32]	Drug delivery (Beva-loaded PLGA NP)	VEGF	Intranasal	Weekly 5	NR	10th; 17th day	~46% at 24th day	10th, 17th and 24th days	BLI
Clavreul et al. [34]	CED (SFN-LNCs)	RTKs (VEGFR-2; VEGRF-3, PDGFR-β, c-kit e Flt-3); Intracellular serine /threonine kinases (Raf-1; B-Raf; B-Raf-mut)	Intratumoral	Single 3.5 µg/mouse	Transcutol^®^HP (0.7 g)	9th day	No reduction at 13th and 16th day	13th and 16th days	MRI; H&E
Sun et al. [51]	Dual-targeting drug delivery (AP1-DOX-NP)	IL-4R	Tail vein	Every other day 10	PBS	10th, 12th, 14th, 16th days	NR	3 and 24 h and 47 days	FLI; H&E; Survival curve
Séhédic et al. [52]	CED (12G5-LNC^188^Re)	CXCR4; Signaling pathways (PI3K/Akt and MAP-kinases); Activation of MMPs; CD11b+ myeloid cells	Intratumoral	Single 10 µL/mouse	Saline	12th day	~100% after 24 days	12th, 17th, 19th to 100 days	MRI; IF; Western Blot; Survival curve
Lu et al. [53]	Drug delivery (CARD-B6)	GBM microenvironment; Transferrin receptors; Telomerase activity	Intravenous	Every other day 0.5 CA4+2.5 DOX+ 0.5 ATRA	PBS	16th to 32nd day	NR	12, 24, 36, 48 h, 30th and 36th day	MRI; LSCA; TUNNEL assay; Survival curve
Xu et al. [54]	Drug delivery (DTX-NPs-dBECM)	GBM microenvironment	Intratumoral	Single 800 µg/mL-DTX+ 10 mg/mL-dBECM	Saline	7th day	~98% at 28th day	7th to 28th days (weekly)	MRI; FLI-ex vivo; H&E; TUNNEL assay; Survival curve
Wang et al. [55]	Systemic therapy (H–S–R NPs1)	Integrin α_v_β_3_ on endothelium; EphA2 Tyrosine kinase receptor on tumor cells and tumor vasculature	Tail vein	Every 2 days NR	Saline	Started when the tumors were visible by BLI	~99% at 12th day	0, 4th, 8th, 12th day; 26th day	BLI; FLI Survival curve;
Lin et al. [56]	Drug delivery (L-BSA NPs)	SPARC; gp60	Tail vein	Daily PTX/ 4-HPR, 2 each	PBS	Started with tumor size (100−200 mm^3^) for 2 days	~93% at 16th day	0, 7 and 16 days; 37th day	BLI; Western Blot; TUNNEL assay; Survival curve
Lin et al. [57]	Systemic therapy (CRLX101)	Topo I inhibition; Hypoxia cascade (CA IX, HIF-1α, VEGF)	NR	Weekly 10	NR	4th and 11th day	NR	20th and 32nd day	H&E; TUNNEL assay; Survival curve
Kuang et al. [58]	Targeted drug delivery (DGL-PEG-T7/shVEGF-DOX)	Transferrin receptor; VEGF gene; Topo II inhibition	Intravenous	Every 2 days 50 µg+8 µg DOX/mouse	Saline	12th, 15th, 18th day	~80% after 18th day	12th and 21st day	BLI; TUNNEL assay; Survival curve
Hu et al. [59]	Dual-targeting drug delivery (AC-NP-PTX)	HSPG; NRP-1	Intravenous	Every 3 days 5	Saline	NR	NR	51st day	FLI; IF; Survival curve
Hu et al. [60]	Dual-targeting drug delivery (MSN-DOX-PDA-NGR)	CD13	Tail vein	Every 3 days 5	Saline	5, 8, 11, 14 days	NR	5th, 10th, 17th, 32nd day	H&E; TUNNEL assay; FLI; MVD; Survival curve
Banerjee et al. [61]	Targeted drug delivery (PSM)	SSTR2	Intravenous	Daily 2	Saline	2 weeks	NR	15th and 36th day	H&E; Survival curve
Zhang et al. [45]	Targeted drug delivery (Functional targeting epirubicin liposomes)	Glut1 on BBB; GBM integrin receptors and neovasculature	Tail vein	Every 3 days 100 µg/Kg	Saline	14th, 18th day	NR	20th and 28th day	FM; Survival curve
Feng et al. [62]	Drug delivery (CooP-NP-PTX)	MDGI (H-FABP/ FABP3)	Intravenous	Every 3 days 5	Saline	2 weeks	NR	47.5	H&E; Survival curve
Costa et al. [44]	Multimodal gene therapy(CTX-coupled SNALPs)	miR-21 (inhibits PDCD4); RhoB; p53; TGF-β; mitochondrial apoptotic networks	Tail vein/ oral	Single 2.5/ 3 days 30	Saline	13th day/ 13th, 14th, 15th day	~45% at 17th day	17th and 30th day	H&E; Western blot; Survival curve
Bechet et al. [63]	Photodynamic therapy (NP-TPC-ATWLPPR)	VEGF receptor; NRP-1	Tail vein	Single 2.8 (1.75 µmol/kg)	NR	NR	~50% after 6 days of iPDT	4th, 6th, 10th day after iPDT	MRI; PET-CT; H&E
Gao et al. [41]	Dual-targeting drug delivery (DTX-IRNPs)	Integrin α_v_β_3_ on endothelium; IL13Rα2	Tail vein	Every 3 days 6	Saline	10th, 11th, 12th day	~71% at 17th day	13tn, 17th, 35th day	IF; H&E; Survavil curve
Wojton et al. [64]	Systemic therapy (SapC-DOPS nanovesicles)	PtdSer	Tail vein	Single 12-SapC 4.6-DOPS	PBS	10th day	NR	11th, 12th, 17th day	IF; H&E; Survival currve
Janic et al. [65]	Cell therapy (FePro)	CD34; AC133; SDF-1-CXCR4 signaling pathway	Intravenous	Single 10 × 10^6^	PBS	12th day	NR	18th day	MRI; Prussian Blue
Day et al. [48]	Photothermal therapy (VEGF-NS)	Integrin α_v_β_3_ on endothelium; VEGFR-2	Tail vein	Single 4.35 × 10^10^ VEGF-NS/mouse	Saline	When tumor reached 3–5 mm	NR	24 h and 3 days after treatment	Intravital microscopy images; H&E; Survival curve
Verreault et al. [43]	Drug delivery (Liposomal Irinophore C^TM^)	GBM microenvironment; GBM vasculature	Intravenous	Weekly 25 Irinophore C^TM^; 15 Caelyx^®^; 2 liposomal vincristine	PBS	21st; 28th; 35th day	~70% at 42nd day	42nd day	H&E
Agemy et al. [66]	Systemic therapy (Iron Oxide Nanoworms- CGKRKD(KLAKLAK)_2_)	Peptides homing to epidermal tumors; GBM vasculature; mitochondrial membrane	Intravenous (U87-MG)	Every other day 5	PBS	3 weeks	NR	5–6 h after the injection	IF; Survival curve
			Intravenous (005)	Every other day 5	PBS	10th, 12th, 14th, 16th, 18th, 20th, 22nd 24th, 26th, 28th, 30th day	NR	5–6 h after the injection	IF; Survival curve
			Intravenous (Sphere)	Every other day 5	PBS	3 weeks	NR	5–6 h after the injection	IF
			Intravenous (Lentiviral)	Every other day 5	PSB	21st, 23th, 25th, 27th, 29th, 31st, 33th, 35th, 37th day	NR	21st, 28th, 35th day	BLI; H&E; IF; Survival curve
Ding et al. [67]	Systemic therapy (P/LLL/AON/Hu/Ms, P/LOEt/AON/Hu/Ms)	Laminin α4 and β1 chain	Intravenous	Single 5	PBS	21st day	~91% NR day	NR	IF; H&E
Hekmatara et al. [68]	Drug delivery (DOX-np)	Endothelial cells	Tail vein	Dayly 1.5	NR	2nd, 5th, 8th day	~100% at 14th day	10th, 14th, 18th day	H&E; IMC
Saito et al. [38]	CED (Ls-TPT)	Topo I inhibition; GBM vasculature	Intratumoral	Single (0.5 mg /mL; 20 μL)	NR	10th day	NR	17th, 19th day	H&E; Survival curve

Abbreviations—NR: Not Reported; Lut: luteolina; Fa-PEG-PCL: folic acid modified poly(ethylene glycol)-poly(e-caprolactone); Beva: Bevacizumab; PLGA: Poly(D,L-lactic-co-glycolic) acid; NP: Nanoparticles; CED: Convection-Enhanced Delivery; SFN: Sorafenib; LNCs: lipid nanocapsules; AP1: CRKRLDRNC peptide; DOX: Doxorubicin; AP1-DOX-NP: tumor homing peptide and DOX-loaded PLA nanoparticles; LNC: Lipid nanocapsules; 12G5: (CD184, #555971, BD Pharmingen); 188Re: Rhenium-188; CARD: nanoparticles with B6 loading three drugs ((CA4+AZO-ATRA+DOX+SPIONs)NPs); B6: peptide motif B6 (1,2-dioleoylsn-glycero-3-phosphoethan-olamine-n-[poly (ethylene glycol)] 2000 (DSPE-PEG2000)); CA4: Combretastin A4; AZO: Azobenzene; ATRA: All-trans retinoic acid; SPIONs: superparamagnetic iron oxide nanocubes; DTX: Docetaxel; DTX-NPs: Docetaxel-loaded nanoparticles; dBECM: decellularized brain extracellular matrix; H-S-R NPs1: heparin-containing NPs with two ligands (SWL and cRGD); L-BSA: LMWP (sequence: CVSRRRRRRGGRRRR)– BSA; BSA: Bovine Serum Albumin; L-BSA NPs: LMWP-modified BSA nanoparticles; LMWP: Low molecular weight protamine; CRLX101: nanoparticle-drug conjugate (NDC), containing approximately 10 wt% CPT conjugated to a linear, cyclodextrin-polyethylene glycol (CD-PEG) copolymer; DGL-PEG: dendrigraft poly-L-lysines; T7: peptide T7 (sequence His-Ala-Lle-Tyr-Pro-Arg-His); shVEGF: inhibition of endogenous VEGF mRNA; DOX: Doxorubicin; AC-NP: ATWLPPR and CGKRK peptide dual-decorated nanoparticulate DDS; PTX: Paclitaxel; MSN: Mesoporous Silica Nanoparticles; PDA: polydopamine; NGR: peptide (CYGGRGNG); MSN-DOX-PDA-NGR: polydopamine (PDA)-coated mesoporous sílica nanoparticles (NPs, MSNs) and the PDA coating was functionalized with Asn-Gly-Arg (NGR); PSM: PTX-loaded SLN modified with TOC; CooP: peptide (ACGLSGLGVA)X; NP-PTX: paclitaxel-loading PEG–PLA nanoparticles; CTX: Chlorotoxin; SNALPs: stable nucleic acid lipid particle; NP-TPC: Nanoparticle- 5-(4-carboxyphenyl)-10,15,20-triphenylchlorin; ATWLPPR: H-Ala-Thr-Trp-Leu-Pro-Pro-Arg-OH; IRNPs: IL-13p and RGD conjugated PEG-PCL nanoparticles; SapC-DOPS: Saposin C-dioleoylphosphatidylserine; FePro: Ferumoxides-Protamine Sulfate; VEGF: Vascular Endothelial Growth Factor; NS: Nanoshells (silica core/gold shell nanoparticles); NWs: iron oxide nanoparticles, dubbed “nanoworms” (NWs); CGKRKD: Cys-Gly-Lys-Arg-Lys peptide; KLAKLAK: α-helical amphipathic peptide D; PMLA (P): poly(β-l-malic acid); LLL: H2N-Leu-Leu-Leu-OH; AON: antisense oligonucleotide; Hu: mAb targeting tumor cells (human TfR); Ms: mAb targeting blood–brain tumor barrier endothelium (mouse TfR); LOEt: H2N-Leu-Leu-Leu-NH2, H2N-Leu-ethylester; TfR: transferrin receptor; DOX-np: Doxorubicin-loaded polysorbate 80-coated poly(butyl cyanoacrylate) (PBCA) nanoparticles; Ls-TPT: liposomal topotecan; RTKs: Receptor Tirosina Kinase; VEGFR-2: Vascular Endothelial Growth Factor Receptor- 2; VEGRF-3: Vascular Endothelial Growth Factor Receptor – 3; PDGFR-β: Platelet derived growth factor receptor- β; c-Kit: Stem Cell Factor Receptor; Flt-3: FMS-like tyrosine kinase 3; Raf-1: Serine/threonine-protein kinase; B-Raf: RAF kinase type B gene; B-Raf-mut: RAF kinase type B gene mutated; IL-4R: interleukin 4 receptor; CXCR4: C-X-C chemokine receptor type 4; PI3K/Akt: signaling pathway; MMPs: Matrix Metalloproteinases; CD11b+: Cluster of Differentiation 11b; GBM: Glioblastoma; EphA2: EPH Receptor A2; SPARC: Secreted Protein, Acidic and Rich in Cysteines; gp60: 60-kDa sialoglycoprotein; CA IX: Carbonic anhydrase IX; HIF-1α: Hypoxia-Inducible Factor 1-α; HSPG: Heparan Sulfate Proteoglycan; NRP-1: Neuropilin-1; CD13: Cluster of Differentiation 13; SSTR2: Somatostatin Receptor Type 2; Glut1: Glucose Transporter 1; BBB: Blood–Brain Barrier; MDGI: Mammary-Derived Growth Inhibitor; H-FABP: Heart-Type Fatty Acid Binding Protein; FABP3: Fatty Acid Binding Protein 3; miR-21: microRNA 21; PDCD4: Programmed Cell Death 4; RhoB: Ras Homolog Family Member B; p53: Protein 53 kDa; TGF-β: Transforming growth factor β; IL13Rα2: Interleukin-13 Receptor α2; PtdSer: Phosphatidylserine; CD34: Cluster of Differentiation 34; AC133: Prominin-1; SDF-1: Stromal Cell-Derived Factor 1; 4-HPR: N-(4-Hydroxyphenyl)retinamide; BLI: Bio-layer interferometry; iPDT: Interstitial Photodynamic Therapy; IHC: Immunohistochemistry; H&E: Eosin and Hematoxylin; MRI: Magnetic Resonance Imaging; IF: Immunofluorescence; WB: Western Blotting; PET-CT: Positron Emission Tomography – Computed Tomography; IMC: Isothermal Microcalorimetry.

**Table 5 ijms-21-04490-t005:** Angiogenic effects evaluation.

Ref.	Angiogenic Markers	Technique Evaluation	Expression of Control Groups	Expression of Treatment Groups	Efficiency of Therapy and Time (d)	Conclusions
Wu et al. [50]	CD31	IHC	Number of microvessels: 37.8 ± 7.3 (NS); 33.4 ± 7.2 (EM)	17.3 ± 5.2 (F-Lut), 11.3 ± 3.1 (Lut-M); 4.1 ± 2.2 (Lut/Fa-PEG-PCL)	~89%/ NR	Lut/Fa-PEG-PCL significantly inhibit the NV of GL261 tumor, play an important role in inhibiting tumor cellular growth
Sousa et al. [32]	VEGF mRNA;	qPCR	8 × 10^−5^ (U87 MG)	2 × 10^−5^ (Beva-loaded PLGA); 1 × 10^−5^(Free Beva)	~49% at 24th day	Beva significantly decrease both extracellular and intracellular VEGF levels, having a higher anti-angiogenic effect compared to the free Beva
	VEGF protein level	ELISA	2000 ng/mL (U87 MG)	1000 ng/mL (Beva-loaded PLGA); 1250 ng/mL (Free Beva)	~38% at 24th day	
Clavreul et al. [34]	CD31	IF	130 ± 9 µm^2^ (HBSS)	124 ± 6 µm^2^ (B-LNC), 128 ± 6 µm^2^ (SFN); 105 ± 5 µm^2^ (SFN-LNC)	~19% at 16th day	SFN-LNCs decreased the proportion of proliferating cells and tumor vessel area, inducing an early increase in tumor blood flow and a vascular normalization process.
	Blood Perfusion	Perfusion MRI	50 ± 3 mL/100 g/min (HBSS)	51 ± 2 mL/100 g/min (B-LNC); 49 ± 3 mL/100 g/ min (SFN); 62 ± 4 mL/100 g/min (SFN-LNC)	~24% (-) at 16th day	
Sun et al. [51]	CD31	IF	NP IF expression < AP1-NP IF expression	NP IF expression < AP1-NP IF expression	NA	AP1-NP has high affinity with vascular endothelial cells.
Séhédic et al. [52]	CD31	IHC	~12.5% (PBS)*	~7.5% (LNC^188^Re)*; ~5% (IgG2a-Re-LNC)*; ~2.5% (12G5-LNC^188^Re)*	~80% at 19th day	The clinical improvement was accompanied by locoregional effects on tumor development including hipovascularization and stimulation of the recruitment of bone marrow derived CD11b- or CD68-positive cells. NOS-II analysis from inside to the external part of the tumor while Arg1 was exclusively present in the peripheral part of the tumor
	CD11b		~1.5% (PBS)*	~8% (LNC^188^Re)*; ~25% (IgG2a-Re-LNC)*; ~22.5% (12G5-LNC^188^Re)*	~93% (-) at 19th day	
	CD68		NA	CD68+/NOSII+ M1	NA	
			NA	CD68+/Arg1+ M2	NA	
	MMP9		NR	NR	NA	
Lu et al. [53]	Blood Perfusion;	LSCI;	1.76 UA (PBS)	1.03 UA (CARD); 1.24 UA (CARD-B6); 1.23 UA (CA4 + ATRA + DOX); 0.74 UA (CARD-B6)	~58% after 16th day	Almost no blood flow existed in the tumor region following treatment with CARD-B6
	CD31	IHC	NR	NR	NA	
Xu et al. [54]	CD31	IHC	100% (Control)	78.1 ± 1.9% (DTX), 58.0 ± 3.9% (DTX-NPs), 30.2 ± 2.8% (DTX-NPs-dBECM)	~70% after 8th day	DTX-NPs-dBECM complex display effective anti-angiogenesis
Wang et al. [55]	CD34+(endothelial lined vessel);	IHC	~45 UA (Control)	~7 UA (H-S); ~20 UA (H-S-R)	~56% at 12th day	H–S–R NPs exerted a significant synergic anti-tumor effect through anti-angiogenic therapy
	CD34-/PAS+ (VM)		~62.33 UA (Control)	~30.67 UA (H-S); ~11.33 UA (H-S-R)	~82% at 12th day	
Lin et al. [56]	CD31, $ SPARC and gp60	IF; WB	NR	NR	NA	Decreased vessel size and number by IHC and reduced CD31 levels by WB in PTX/4-HPR treatment group
Lin et al. [57]	CD31	IHC	Control	> CRLX101 and CPT	NA	In vivo results indicate that CRLX101 was more effective than CPT in inducing apoptosis and suppressing angiogenesis due to CRLX101′s improved drug delivery profile and enhanced permeability and retention effect
	CA IX		NR	CRLX101 > CPT	NA	
	VEGF	IHC; WB	1.0 (Vehicle)	~0.1 CRLX101 ~0.6 CPT	~40% at 14th day	
Kuang et al. [58]	CD34/Lectin#	IF	NR	NR	NA	DGL-PEG-T7/shVEGF could inhibit VEGF mRNA much better than DGL-PEG/shVEGF. This could be explained as the nanoparticles bind to TfR on the surface of the tumor cells via the T7 peptide. shVEGF and DOX delivered by DGL-PEG-T7 could inhibit tumor growth and angiogenesis
	VEGF mRNA;	RT-PCR	100% Saline	69.2% (DGL-PEG/shVEGF); 41.6% (DLG-PEG-T7/shVEGF); 49.0% DGL-PEG-T7/shVEGF-DOX	~51% at 21st day	
Hu et al. [59]	CD31	IF	NR	NR	NA	The abundant extracellular matrix-derived HSPG and enhanced tumor penetration ability mediated by NRP-1 protein, allowed the AC-NP to achieve angiogenic blood vessels and tumor microenvironment with dual-targeting effect.
Hu et al. [60]	CD31	IHC	NR	NR	NA	Delivered the drugs into the glioma cells was more efficiently, induced more cell apoptosis and necrosis with fewer MV in the MSN-DOX-PDA-NGR group
Banerjee et al. [61]	CD31 (MV density)	IHC	~180 UA	~150 UA (Taxol); ~90 UA (PSLN); ~25 UA (PSM)	~86% at 15th day	PSM holds high potential dual-targeting for tumor neovasculature and tumor cells due to TOC (in PSM surface) interaction with SSTR2 expressed in EC NV, PTX improves AA effects when encapsuladed.
Zhang et al. [45]	CD3, DiI	IF	NR	NR	NA	FTEL are able to destroy brain glioblastoma NV and to extend the survival of brain glioblastoma-bearing mice
Feng et al. [62]	CD31	IF	~97% (Taxol)	~65% (NP-PTX); ~30% (CooP-NP-PTX)	~69% at 1 week after treatment	CooP-NP-PTX led to an effective tumor angiogenic blood vessel and glioma cell, holds great potential to improve anticancer activity and avoid the drawbacks of anti-angiogenic therapy alone.
Costa et al. [44]	CD31	IHC	145 ± 63 cells	113 ± 79 cells (Mismatch + Sunitinib); 88 ± 69 cells/ (Anti-miRNA-21 + Sunitinib)	~39% at 17th day	CTX-coupled SNALP formulated anti-miR-21 OG reduction of the number of vascular EC
Bechet et al. [63]	VEGF	IHC	NR	NR	NA	Vascular disruption and edema into both tumor and BAT areas; Intense decrease of VEGF expression after iPDT
Gao et al. [41]	HIF1α	IF	Low HIF1α expression (Saline)	HIF1α expression (DTX-ILNPs)>(DTX-RNPs)	NR (+) at 17th day	DTX-ILNPs increased the expression of HIF1a in tumor and could be effectively for antiangiogenesis problems
Wojton et al. [64]	CD31	IF	NR	NR	NA	SapC-DOPS targets glioma cells (DAPI) and tumor vasculature (CD31), but not normal brain tissue.
Janic et al. [65]	CD31, vWF	IHC	NR	NR	NA	Strong expression of vWF and CD31 in iron-labeled CB AC 133+ EPC positive cells overlapped with tumor vasculature
Day et al. [48]	Vessel morphology	Intravital microscopy; H&E	Increase of 18% of VD (Saline)	Decreade of 24% of VD (VEGF-NSs)	~42% after 3 days of treatment	Treatment with VEGF-NS, following laser exposure disrupts tumor vessels, majorly in tumor and at its periphery, but not in the adjacent normal brain. (Intravital microscopy); vessel dilation and hemorrhaging within the tumor exposed to VEGF-NSs and PEG-NSs (H&E).
Verreault et al. [43]	CD31, Collagen IV, NG2	IHC, IF	Collagen IV-free CD31: ~12 pixels (Control tumor)	Collagen IV-free CD31: ~9.5 pixels (Irinophore C^TM^, Caelyx^®^)	~21% at 42nd day	Irinophore C^TM^ restored the BMA and reduced BVD of the tumor vasculature, suggesting a restoration of the vessel architecture to a more normal state. In addition, it increased the quantity of vessel staining in the center of tumors, suggesting a more homogenous distribution of blood across the entire tumor, as well as reduced K trans values. No changes in ECD in the TTA or the periphery of tumors treated with Caelyx^®^ or liposomal vincristine
			BVD: ~11 pixels (Control tumor)	BVD: ~6.5 pixels (Irinophore C^TM^, Caelyx^®^)	~39% at 42nd day	
			NG2-free CD31: ~2.5 pixels (Control tumor)	NG2-free CD31: ~0.75 pixels (Irinophore C^TM^, Caelyx^®^)	70% at 42nd day	
			CD31-free Collagen IV: ~0.9 pixels (Control tumor)	CD31-free Collagen IV: ~0.9 pixels (Irinophore C^TM^, Caelyx^®^)	~0% at 42nd day	
	Ktrans	DCE-MRI	0.0232 mL/g/min (Control)	0.0034 mL/g/min (Irinophore C^TM^)	~85% at 42nd day	
Agemy et al. [66]	CD31	IF	NR	NR	NA	NWs coinjected with iRGD had spread into the extravascular tumor tissue, whereas NWs coinjected with CRGDC mainly accumulated in tumor vessels; Vascular structures were filled with CGKRKD (KLAKLAK)_2_-NWs; destruction of the BV by the NWs
Ding et al. [67]	Laminin 411 (α1 and β4 chains)	IHC	Vessel area: 5.5% (PBS)	Vessel area: 3.75% (LOEt); 2.5% (LLL)	~55% after 21st day	Antitumor efficacy of LOEt and LLL due to reduced production of laminin-411 chains and decreased angiogenesis
Hekmatara et al. [68]	VEGF;	IHC	NR	2 score (Dox-sol); 1 score (Dox-np)	NA at 18th day	Dox-sol led to a slight decrease of necrosis and MVP whereas Dox-np drastically decreases necrosis and led to the complete disappearance of MVP.
	Isolectin B4		NR	~7% (Dox-sol); ~1%(Dos-np)	~86% at 18th day	
	Microvascular proliferation:	H&E	NR	1 score (Dox-sol); 0 score (Dox-np)	NA at 18th day	
Saito et al. [38]	Laminin	IHC; WB	BV: control ≈ free-TPT	BV: Ls-TPT< free-TPT	NA at 14th day	Marked decrease in blood vessels in Ls-TPT group, as well as hypophosphoriylated Akt, whereas control and free TPT shower high density of blood vessels
	p-Akt	WB	NR	Ls-TPT< free-TPT or control	NA at 14th day	

**Abbreviations**—Ref: Reference; IHC: Immunohistochemistry; NS: Normal Saline group; EM: Blank microparticle group; MV: microvessels; NV: neovascularization; NT: Neonatal tumor; F-lut: Free luteolin group; Lut-M: luteolin/MPEG-PCL nanoparticle group; Fa-Lut: luteolin/Fa-PEG-PCL nanoparticle group; qPCR: Quantitative Reverse transcription polymerase chain reaction; ELISA: Enzyme Linked Immuno Sorbent Assay; Beva: Bevacizumab; Beva-loaded PLGA NP: Bevacizumab loaded PLGA nanoparticles; VEGF: Vascular endothelial growth factor; IF: Immunofluorescence; LSCI: laser speckle contrast images; SFN-LNC: sorafenib-loaded lipid nanocapsules; HBSS: Hank’s Balanced Salt Solution; B-LNC: Blank lipid nanocapsules; SFN: sorafenib; DDS: Drug delivery system; LNC^188^Re: Rhenium-188 loaded in the core of a lipid nanocapsule; 12G5: function blocking antibody directed at CXCR4; CARD-B6: NPs with B6 loading three drugs ^(CA4+AZO-ATRA+DOX+SPIONs)^NPs-B6; CA4: Combretastin A4; ATRA: All-trans retinoic acid; AZO: Azobenzene; B6: 1,2-dioleoylsn-glycero-3-phosphoethanolamine-n-[poly (ethylene glycol)] 2000; SPIONs: superparamagnetic iron oxide nanocubes; dBECM: decellularized brain extracellular matrix; DTX: Docetaxel; DTX-NPs: Docetaxel-loaded nanoparticles; PAS: VM: Vasculogenic mimicry; H-S-R: heparin-containing polymer SWL and cRGD; H-S:; WB: Western blot; PTX/4HPR; CA IX: Carbone anhydrase IX; CPT: Camptothecin; AC-NP: ATWLPPR and CGKRK peptide dual-decorated nanoparticulate drug delivery system; MSN: mesoporous silica nanoparticles; DOX: Doxorubicin; PDA: polydopamine; NGR: Asn-Gly-Arg; AA: Antiangiogenic; EC: endothelial cells; NV: Neovasculature; TOC: Tyr-3-octreotide; PSM: PTX-loaded SLN modified with TOC; SSTR2: Somatostatin receptor 2; PSLN: PTX: Paclitaxel; SLN: Solid lipid nanoparticle; FTEL: functional targeting epirubicin liposomes; OG: Oligonucleotides; CTX: chlorotoxin; SNALP: stable nucleic acid lipid particle; BAT: Brain adjacent tumor; iPDT: Interstitial photodynamic therapy; SAPC-Dops: SAPCSaposin C-dioleoylphosphatidylserine; DAPI: CB AC133+ EPC: Cord blood AC133+ endothelial progenitors cells; VEGF-NSs: VEGF-coated nanoshells; PEG-NSs: poly(ethylene glycol)-coated nanoshells; VD: Vessel density; ECD: EC density; TTA: Total tumor área; BV: Blood vessel; BMA: Basement membrane architecture; NWs: iron oxide nanoparticles, dubbed “nanoworms” (NWs); CGKRKD: Cys-Gly-Lys-Arg-Lys peptide; KLAKLAK: α-helical amphipathic peptide D; iRGD: a tumor-penetrating peptide; MVP: Microvascular proliferation; Dox-sol: doxorubicin in solution; Dox-np: doxorubicin bound to polysorbate 80-coated poly(butyl cyanoacrylate) nanoparticles; TPT: Topotecan; Ls-TPT: topotecan; p-Akt: phosphoriylated Akt. Note—* ratio on total tumor area; CD68+/NOSII+ M1: phenotype cells notably associated with tumor destruction and tissue damage; CD68+/Arg1+ M2: phenotype usually associated with tumor promotion and tissue remodeling; #Functional blood vessels evaluation by lectin marker; FTEL: functional targeting epirubicin liposomes; ktrans: a volume transfer constant of a solute between the blood vessels and extra-cellular tissue compartment; Dynamic Contrast Enhanced (DCE)-MRI; $ SPARC and gp60 overexpression was found on glioma and tumor vessel endothelium, exploring the use in brain-targeting biomimetic delivery; Microvascular proliferation: 0—no microvascular proliferation, 1—solitary nodules of microvascular proliferation, 2—more than five nodules of microvascular proliferation; VEGF: 0—no positive cells, 1—weak staining intensity, 2—moderate staining intensity, 3—strong staining intensity.

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
