# Peer review of "Antiangiogenic Targets for Glioblastoma Therapy from a Pre-Clinical Approach, Using Nanoformulations"

_ijms, 2020, doi:10.3390/ijms21124490_

Round 1
Reviewer 1 Report
The authors present a systematic review in which they summarize several aspct of Glioblastoma, including tumor cell characterization, GBM animal models, nanoformulation composition, and finally the therapeutic anti-angiogenic process. This work represents a significant contribution to the field of anti-angiogenic approach for glioblastoma therapy. Data are good represented, and the tables, although they contain many data, are well presented and easy to read. In addition, the references reported are appropriate, updated.
Minor comments:
- The authors, as a completion of this interesting work, should add a paragraph about clinical trials involving patients affected by GBM and treated with nanoformulations.
- I would suggest to chenge the title, adding the words glioblastoma, anti-angiogenic therapy and nanoformulation or nanoparticles, taking up what reported in the review.
- At page 14, line 299, a reference is missing in the description of the 6 studies “in which functionalized nanoparticles with different molecules were used such as proteins, peptides, and tumor markers”
- At page 22, line 590, the authors should correct GB with GBM
- in the same page, at line 592 the authors missed to add the number for the reference Danhier et al., 2015.
In summary, this review contains an interesting description of current knowledge on GBM cell and animal models and the effects of nanoformulations on various processes related to GBM angiogenesis. The manuscript is of interest, and it is well written.
Author Response
June 07, 2020
ijms-822283
Antiangiogenic targets for Glioblastoma therapy from a pre-clinical approach, using nanoformulations
To
Assistant Editor: Tammy Lv
International Journal of Molecular Sciences
Dear Editor,
We are sending the revised version of the manuscript entitled, “Antiangiogenic targets for Glioblastoma therapy from a pre-clinical approach, using nanoformulations”, Manuscript ijms-822283, with point-by-point corrections (see below) suggested by the reviewer 1. The changes of the manuscript were highlighted using the “Track Changes” option.
Thank you again for your time and consideration. We hope the paper is now suitable for publication in International Journal of Molecular Sciences. We are looking forward to hearing your decision.
Sincerely,
Lionel Gamarra
Reviewer #1
Comments and Suggestions for Authors
The authors present a systematic review in which they summarize several aspect of Glioblastoma, including tumor cell characterization, GBM animal models, nanoformulation composition, and finally the therapeutic anti-angiogenic process. This work represents a significant contribution to the field of anti-angiogenic approach for glioblastoma therapy. Data are good represented, and the tables, although they contain many data, are well presented and easy to read. In addition, the references reported are appropriate, updated.
Minor comments:
Question 1 - The authors, as a completion of this interesting work, should add a paragraph about clinical trials involving patients affected by GBM and treated with nanoformulations.
Answer: Thank you for your suggestion. We added this information requested by the reviewer in the discussion section of the manuscript, although, until now, only one type of nanoparticle, a polymer of carmustine wafers is used in clinical practice for glioblastoma patients. Other nanoformulations for glioblastoma treatment represent an emergent alternative treatment, but not yet have clear evidence in clinical trials on antiangiogenic results. In addition, the proposal of this systematic review was focused only on pre-clinical research, as highlighted in the title, aiming the quantitative and qualitative analysis of the best conditions on nanoformulations for antiangiogenic therapy, but the nanoformulations and angiogenic targets variability, associated to inconclusive methods for antiangiogenic therapy evaluation did not allow to conclude what the best antiangiogenic therapy for glioblastoma in pre-clinical research.
Question 2 - I would suggest to change the title, adding the words glioblastoma, anti-angiogenic therapy and nanoformulation or nanoparticles, taking up what reported in the review.
Answer: Thank you for your suggestion. We added in the title the word nanoformulation as requested by the Reviewer, changing the title of the manuscript to “Antiangiogenic targets for Glioblastoma therapy from a pre-clinical approach, using nanoformulations”
Question 3 - At page 14, line 299, a reference is missing in the description of the 6 studies “in which functionalized nanoparticles with different molecules were used such as proteins, peptides, and tumor markers”
Answer: Thank you for your observations. Sorry, the correct number is five studies (18%) and not six, as described wrongly in the manuscript that used the systemic delivery system. So, the reference is correct, the number of studies was mistaken.
Question 4 - At page 22, line 590, the authors should correct GB with GBM
Answer: Thank you for your observation. We corrected the abbreviation "GBM" appointed in of the manuscript (line 590 on page 22), and reviewed all abbreviations of the manuscript
Question 5 - in the same page, at line 592 the authors missed to add the number for the reference Danhier et al., 2015.
Answer: Thank you for your observation. We corrected the reference Danhier et al., 2015 that corresponds currently to reference 89 of the manuscript.
In summary, this review contains an interesting description of current knowledge on GBM cell and animal models and the effects of nanoformulations on various processes related to GBM angiogenesis. The manuscript is of interest, and it is well written.
Reviewer 2 Report
This systematic review by Gabriel N. A. Rego, et al. is a comprehensive and state-of-the-art presentation of the new insight into Antiangiogenic targets for Glioblastoma therapy from a pre-clinical approach. Their approach entails a new and substantial contribution to current literature on the subject matter; however, I have the following minor but serious concerns:
- This review study is largely confirmatory of a previously published study by Clavreul A et al., Int J Nanomedicine. 2019 Apr 9;14:2497-2513.; Arrillaga-Romany I et al., CNS Oncol. 2014;3(5):349-58.; Gerstner ER et al., Cancer J. 2012 Jan-Feb;18(1):45-50, Preclinical studies underscore the importance of neovascularization for tumor survival, making angiogenesis an important treatment target in GBM and therefore lacks significant novelty.
- This review is too long and, in many places repetitive. Authors should review the paper content for redundancy and ensure only essentials are left. Please reduce the paper about 15-20 pages.
- Though there is not standard limit for references in review articles, i feel over 131 references is just too many. Authors should look at cutting this down to < 100.
- Please focus and discuss some paragraphs of novel treatment approach of Glioblastoma by targeting vasculature normalization.
- Please also discuss the targeting tumor angiogenesis with plant-derived natural products regarding the emerging trends in Glioblastoma cancer therapy
- There are also a few errors in English language grammar that require the authors' attention.
- All abbreviations must be defined when they are first used.
- Authors may want to provide a more representative Graphical Abstract
This manuscript cannot be accepted in its present form, I recommend REJECTION.
Author Response
Reviewer #2
This systematic review by Gabriel N. A. Rego, et al. is a comprehensive and state-of-the-art presentation of the new insight into Antiangiogenic targets for Glioblastoma therapy from a pre-clinical approach. Their approach entails a new and substantial contribution to current literature on the subject matter; however, I have the following minor but serious concerns:
1-This review study is largely confirmatory of a previously published study by Clavreul A et al., Int J Nanomedicine. 2019 Apr 9;14:2497-2513.; Arrillaga-Romany I et al., CNS Oncol. 2014;3(5):349-58.; Gerstner ER et al., Cancer J. 2012 Jan-Feb;18(1):45-50, Preclinical studies underscore the importance of neovascularization for tumor survival, making angiogenesis an important treatment target in GBM and therefore lacks significant novelty.
Answer: Thank you for your comments, but we would like to highlight some observations about your comments. Firstly, the studies cited by the reviewer were all narrative reviews, which differ from our manuscript that is a systematic review that follows objective criteria (without bias) for the selection of articles included in the study (PRISMA guideline), however, in the narrative review, there is no scientific rigor in the selection, so much so that only one article selected by the systematic review coincides with the other articles mentioned by the reviewer.
Secondly, all the studies included in our review analyzed the antiangiogenic therapeutic effects of different strategies having in common the use of nanoformulations for drug delivery in GBM orthotopic models developed in rodents. In addition, we performed quantifications of these effects on markers related to the angiogenic process.
Differently, the study by Clavreul A. et al. (2019) through a narrative review, proposed the analysis of the nonviral delivery methods used to increase the activity of antiangiogenic factors, focusing mainly in the survival curve evaluation of intracranial animal models of GB after therapy, which represents an unspecific aspect of the real the antiangiogenic effect. So, this review described in a general way the aspects involved in glioblastoma therapy applied in the orthotopic model.
The study by Arrillaga-Romany I et al. (2014) makes a brief narrative review of antiangiogenic agents for the treatment of glioblastoma in clinical research, without the use of nanoformulations, focusing mainly on the action of the drug Bevacizumab and its angiogenesis targets, as well as the study by Gerstner ER et al. (2012) that reported briefly the overview of antiangiogenic therapy for GBM in clinical research, highlighting the angiogenic agents, toxicities of antiangiogenic agents as a narrative review.
2-This review is too long and, in many places repetitive. Authors should review the paper content for redundancy and ensure only essentials are left. Please reduce the paper about 15-20 pages.
Answer: Thank you for your suggestion. Considering that it is a systematic review and that requires details of all the data collected, the manuscript itself becomes more extensive and as we find great variability in the main aspects analyzed in the included studies, this made the description longer. In addition, almost 24 pages represent only tables, figures, and references, it is impossible to reduce the manuscript from 15 to 20 pages. Nonetheless, we seek to reduce to the maximum the information replicates and redundancies, maintaining only what we deem essential, and adding only the requests made by the reviewers.
3-Though there is not standard limit for references in review articles, I feel over 131 references is just too many. Authors should look at cutting this down to < 100.
Answer: Thank you for your suggestion. We agreed with your suggestion and reduced the number of references as much as possible, concluding the manuscript with 104 references.
4-Please focus and discuss some paragraphs of novel treatment approach of Glioblastoma by targeting vasculature normalization.
Answer: Thank you for your comments. We added more information about this issue in the discussion section of the manuscript.
5-Please also discuss the targeting tumor angiogenesis with plant-derived natural products regarding the emerging trends in Glioblastoma cancer therapy
Answer: Thank you for your observation. We added more information on this issue, in which 78% of the selected studies used drugs in the nanoformulation, 52% of these drugs are derived from different types of plants. The taxanes (Paclitaxel and Docetaxel) were more representative in 36 and 18% of drugs used in the studies, respectively. Some of these were used in the nanoformulations conjugated with other drugs derived or not from plants, as well as a coadjuvant of radiotherapy due to its radiation sensitized action.
6-There are also a few errors in English language grammar that require the authors' attention.
Answer: Thank you for your observation. Before the manuscript submission, the English language of the manuscript was reviewed by the MDPI English editing service, and due to the reviewer's comments appointing errors in English language grammar, we requested a new review by the same service. So, the manuscript was reviewed again by the MDPI English editing service.
7-All abbreviations must be defined when they are first used.
Answer: Thank you for your suggestion. We reviewed again all abbreviations of the manuscript.
8- Authors may want to provide a more representative Graphical Abstract
Answer: Thank you for your suggestion. We modified the graphical abstract improving the representative way of the review results
Reviewer 3 Report
Rego G and co-authors publish a very detailed review on the different approaches of anti-angiogenic treatment use in pre-clinical models.
This is a very interesting review, many details are given all along the text.
I will accept it if the authors are adding a figure in which they sum up the consequences of anti-angiogenic treatment on the cell signaling, as described in conclusion.
Author Response
Reviewer #3
Rego G and co-authors publish a very detailed review on the different approaches of anti-angiogenic treatment use in pre-clinical models.
This is a very interesting review, many details are given all along the text.
I will accept it if the authors are adding a figure in which they sum up the consequences of anti-angiogenic treatment on the cell signaling, as described in conclusion.
Answer: Thank you for your observation. We added a Figure at the end of the results section of the manuscript, highlighting the main results found on the systematic review for data analysis on nanoformulations, antiangiogenic agents, tumor cells, animals used in studies, therapy targets, angiogenic markers and antiangiogenic efficiency found in each study, accordingly to angiogenic markers used in the antiangiogenic evaluation. The reviewer's suggestion to add a figure with the results of the review enriched the manuscript because the data found in the review became more evident, which will facilitate the reader regarding the analysis of the data found.
Round 2
Reviewer 2 Report
This is a revised version of the manuscript previously submitted by Gabriel et al. globally the authors have addressed the majority of the previously raised comments although some questions remain as indicated below.
- Please focus and discuss some paragraphs of novel treatment approach of Glioblastoma by targeting vasculature normalization.
- Please also discuss the targeting tumor angiogenesis with plant-derived natural products regarding the emerging trends in Glioblastoma cancer therapy.
Author Response
Reviewer #2
This is a revised version of the manuscript previously submitted by Gabriel et al. globally the authors have addressed the majority of the previously raised comments although some questions remain as indicated below.
- Please focus and discuss some paragraphs of novel treatment approach of Glioblastoma by targeting vasculature normalization.
Answer: Thank you for your comments. The discussion regarding the treatment approach of Glioblastoma by targeting vasculature normalization was improved in the discussion section of the manuscript according to the reviewer’s request.
- Please also discuss the targeting tumor angiogenesis with plant-derived natural products regarding the emerging trends in Glioblastoma cancer therapy.
Answer: Thank you for your comments. we added in the discussion section of the manuscript more information about targeting tumor angiogenesis with plant-derived natural products and the emerging trends in GBM therapy